# Optimistic Dynamic Regret Bounds

## Abstract

Online Learning (OL) algorithms have originally been developed to guarantee good performances when comparing their output to the best fixed strategy. The question of performance with respect to dynamic strategies remains an active research topic. We develop in this work dynamic adaptations of classical OL algorithms based on the use of experts' advice and the notion of optimism. We also propose a constructivist method to generate those advices and eventually provide both theoretical and experimental guarantees for our procedures.

## 1 Introduction

Online learning (OL) is a paradigm in which data is processed sequentially, either because the practitionner does not collect all data prior to analysis or because the dataset dyamically evolves through time, or simply because handling batch of data is numerically too demanding. From the seminal work of Zinkevich (2003), which proposed an online version of the celebrated gradient descent algorithm, OL has been at the core of many contributions (we refer to Hazan et al., 2007; Duchi et al., 2011; Rakhlin and Sridharan, 2013a for an overview). The classical performance criterion of an online learning algorithm is the *static regret*. Given a sequence of loss functions $(\ell_t : \mathcal{K} \to \mathbb{R})_{t \geq 1}$, the static regret compares the efficiency of a sequence of predictors $\hat{\mu} = (\hat{\mu}_t)_{t \geq 1}$ to the best fixed strategy. Its definition is, for any horizon $T > 0$

$$\text{S-Regret}_T(\hat{\mu}) = \sum_{t=1}^{T} \ell_t(\hat{\mu}_t) - \inf_{\mu_0 \in \mathcal{K}} \sum_{t=1}^{T} \ell_t(\mu_0).$$

Classical upper bounds on static regret involve a sub-linear rate. For instance, Zinkevich (2003) proposed a $\mathcal{O}(\sqrt{T})$ bound for Online Gradient Descent (OGD) which is valid for convex losses. Hazan et al. (2007) proved a $\mathcal{O}(d \log(T))$ rate for the Online Newton Step (ONS) algorithm with exp-concave losses when $\mathcal{K} \subseteq \mathbb{R}^d$.

**Dynamic Regret.** Static regret may not be sufficient to assert the efficiency of an online algorithm as the class of static strategies is limited compared to all possible strategies. Hence the notion of *dynamic regret* introduced by Zinkevich (2003) and further developed by Hall and Willett (2013). For any sequence $\hat{\mu}$ of predictors and any sequence $\mu$ of dynamic strategies, the dynamic regret is given by

$$\text{D-Regret}_T(\hat{\mu}, \mu) = \sum_{t=1}^{T} \ell_t(\hat{\mu}_t) - \sum_{t=1}^{T} \ell_t(\mu_t).$$

Dynamic regret has attracteed many studies recently, especially when the comparator sequence is $\mu = \mu^* := (\inf_{\mu \in \mathcal{K}} \ell_t(\mu))_{t \geq 1}$ (worst-case dynamic regret, as in Besbes et al., 2015; Jadbabaie et al., 2015; Yang et al., 2016; Zhang et al., 2017; 2018b; Zhao and Zhang, 2021) but also for any comparator sequence (universal dynamic regret, as in Zhao et al., 2020). Those works have established various upper bounds which depend on measures of the cumulative distance between successive optima. Historically, Zinkevich (2003) introduced the *path length* to measure this discrepancy: for all $T$, for any sequence $\mu = (\mu_t)_{t \geq 0}$,

$$P_T(\mu) = \sum_{t=0}^{T-1} \|\mu_{t+1} - \mu_t\|.$$

Zhang et al. (2017) introduced the *squared path length*: for all $T$, for any sequence $\mu = (\mu_t)_{t \geq 0}$,

$$S_T(\mu) = \sum_{t=0}^{T-1} \|\mu_{t+1} - \mu_t\|^2.$$

The *function variation* has been introduced by Besbes et al. (2015): for all $T$, and any sequence of losses $(\ell_t)_{t \geq 0}$ (these are provided by the environment),

$$V_T^\ell(\mu) = \sum_{t=0}^{T-1} \sup_{\mu \in \mathcal{K}} \|\ell_{t+1}(\mu) - \ell_t(\mu)\|.$$

When using the path length[1] $P_T^* := P_T(\mu^*)$ of the minimisers $\mu^* = (\mu_t^*)_{t \geq 0}$, dynamic regret of OGD is at most $\mathcal{O}(\sqrt{T(1 + P_T^*)})$ for convex functions (Zinkevich, 2003; Yang et al., 2016). We similarly define $S_T^* := S_T(\mu^*)$.

For strongly convex and smooth functions, Mokhtari et al. (2016) established that the dynamic regret is $\mathcal{O}(P_T^*)$. Zhang et al. (2017) introduced the Online Multiple Gradient Descent (OMGD) and the Online Multiple Newton Update (OMNU) which achieved a $\mathcal{O}(\min(P_T^*, S_T^*))$ dynamic regret. Yang et al. (2016) showed that the $\mathcal{O}(P_T^*)$ rate is also reached for convex and smooth functions under the assumption that all minimisers lie onto the interior of a convex set of interest. Besbes et al. (2015) proved a $\mathcal{O}(T^{2/3}(V_T^*)^{1/3})$ dynamic regret for OGD with a restarting strategy. Finally, Baby and Wang (2019) improved the rate to $\mathcal{O}(T^{1/3}(V_T^*)^{2/3})$ for 1-dimensional square loss with filtering techniques. Note that all the aforementioned results assume implicitly access to $P_T^*$, $S_T^*$, $V_T^*$. We note that a notion of *universal dynamic regret* has been studied by Zhang et al. (2018a); Zhao et al. (2020; 2022) to compete with any $P_T(\mu)$, $S_T(\mu)$, $V_T(\mu)$ rather than $P_T^*$, $S_T^*$, $V_T^*$.

**Optimistic online learning.** This refers to a subfield of online learning which exploits, at each time step, a (possibly) history-dependent additional information provided by an expert. Being optimistic in this context is relying on the fact that the experts' advice is relevant and can be exploited within an optimisation procedure. Optimistic online learning can be traced back to Hazan and Kale (2010); Chiang et al. (2012) and has been further developed by Rakhlin and Sridharan (2013a;b) which introduced the celebrated Optimistic Mirror Descent (OMD). Those works involved static regret bound exploiting explicitly the experts' advice. Jadbabaie et al. (2015) bridged the gap between dynamic regret and optimistic online learning by providing an adaptive version of OMD alllowing to obtain dynamic regret bounds for bounded convex functions.

## 1.1 Contributions and outline

Our work is in line with the framework of Jadbabaie et al. (2015). We propose new optimistic algorithms for strongly convex functions, with their associated dynamic worst-case regret bounds. Our performance is compared to the best possible predictors $\mu^*$. First, we establish a procedure named ADJUST (see Sec. 2) which takes as input a candidate predictor (for instance the one generated by classical OGD) and we adjust its trajectory with regards to an experts' advice. Second, we propose an algorithm to construct a sequence of experts' advice (CONSTRUCT) which takes inspiration from the OMGD algorithm of Zhang et al. (2017).

Using ADJUST, we provide updated versions of three classical online algorithms: the Online Gradient Descent (OGD, Zinkevich, 2003), the Online Newton Step (ONS, Hazan et al., 2007) and AdaGrad (Duchi et al., 2011). Those updated versions allow to adapt S-Regret proofs of Hazan (2019) to D-Regret proofs. This leads to D-Regret worst-case guarantees that hold for strongly convex losses: in particular the losses are not necessarily smooth. This focus on non-smooth losses is a new setting of interest in the dynamic regret field and has been recently studied by Baby and Wang (2022). Note that our guarantees hold for any expert advices satisfying technical conditions (notably satisfied by CONSTRUCT).

---

[1]similar definitions hold for the squared path length and the function variation.

More precisely, we present fully empirical D-Regret bounds for a specific choice of computable experts $\nu$ (detailed in Sec. 3) which depend on $P_T(\nu)$, $S_T(\nu)$. The fact that it does not depend on $P_T(\mu^*)$, $S_T(\mu^*)$ is remarkable as we do not need to know the true minimisers to reach an empirical upper bound. Our D-Regret bounds have the following form:

$$\text{D-Regret}_T(\hat{\mu}, \mu^*) \leq f\left(P_T(\nu), S_T(\nu)\right) + g(T).$$

Our main results are gathered in Thms. 3.1, 3.3 and 3.5. A key takeaway message is that we decorrelate the impact of the time horizon $T$ from the impact of the path lengths $P_T$, $S_T$. Our bounds feature a sum of two terms: a function $g(T)$ and a function $f(P_T, S_T)$ combining the different paths. Those results differ from the (optimal) state-of-the-art bound for convex functions of Zhang et al. (2018a, Theorem 4) which is in $\mathcal{O}(\sqrt{T(1 + P_T)})$. Such a decoupling allows to pin down more precisely what is costful in the learning process, be it the optimisation phase or the complexity of the problem.

Additionally to classical D-Regret bounds on the sequence of losses $(\ell_t)_{t \geq 1}$, our updated OGD, ONS and AdaGrad provably satisfy dynamic regret bounds on the loss sequence defined for any $t \geq 1$: $\mathbb{E}_{t-1}[\ell_t] = \mathbb{E}[\ell_t \mid \mathcal{F}_{t-1}]$ with $(\mathcal{F}_t)_{t \geq 1}$ a filtration adapted to the environment and $\mathbb{E}_{t-1}[\ell_t]$. This ensures that our predictors are robust to the randomness of the environment. Thus, we define the *Dynamic Cumulative Risk* (D-C-Risk) as follows: for any predictable[2] sequence $\hat{\mu}$ of predictors (i.e., $\hat{\mu}_i$ is $\mathcal{F}_{i-1}$ measurable) and sequence $\mu$ of dynamic strategies, we denote $L_t = \mathbb{E}_{t-1}[\ell_t]$,

$$\text{D-C-Risk}_T(\hat{\mu}, \mu) = \sum_{t=1}^{T} L_t(\hat{\mu}_t) - \sum_{t=1}^{T} L_t(\mu_t).$$

We then obtain for our updated OGD, ONS and AdaGrad, dynamic cumulative risks of the following form: for any predictable sequences $\hat{\mu}$ and $\mu$, any experts' advice $\nu$, with probability at least $1 - \delta$,

$$\text{D-C-Risk}_T(\hat{\mu}, \mu) \leq f(P_T(\nu), S_T(\nu)) + g(T, \delta).$$

Those results, are universal in the sense that our comparators can be any predictable sequence (and not the true minimisers) and pessimistic as the bound does not involves those comparators. They are gathered in Thms. 3.2, 3.4 and 3.6.

We perform experiments (Sec. 4) to assess our algorithms efficiency. In particular, we test one of our methods (Dynamic OGD) on several real-life datasets to compare its performance to OGD or OMGD. The comparison with OMGD is particularly relevant since our theoretical results, while slightly weaker than Zhang et al. (2017, Corollary 4), have a broader range of application, and require weaker assumptions. Finally, we propose a toy experiment which illustrates the interest of dynamic cumulative risks as performance criterion for noisy learning problems, and we show that the true minimisers are not always the good objective to target when focusing on the D-C-Risk.

We close the paper with some additional technical background (Appendix A), further details on motivation (Appendix C), and we defer to Appendices D and E the proofs of the results of Sec. 3.

## 2  A new optimistic auxiliary procedure

**Framework.**  In this work (unless precised explicitly), we use the following mathematical objects and their associated assumptions:

- The set of predictors $\mathcal{K} \subseteq \mathbb{R}^d$ is a closed convex set with finite diameter $D$.

---

[2]in the sense that predictors only depend on the past.

- We denote by $||.||$ the Euclidean norm on $\mathbb{R}^d$. Our loss functions $(\ell_t)_{t\geq 1}$ are $\lambda$-strongly convex:

$$\forall (t, \mu, \mu_0) \in \mathbb{N}/\{0\} \times \mathcal{K}^2, \ell_t(\mu) - \ell_t(\mu_0) \leq \langle \nabla \ell_t(\mu), \mu - \mu_0 \rangle - \lambda \|\mu - \mu_0\|^2.$$

- All gradients are bounded by some constant $G$: $\forall t \geq 1, \mu \in \mathcal{K}, ||\nabla \ell_t(\mu)|| \leq G$.

## 2.1 The Adjust algorithm

We introduce an optimistic procedure (namely ADJUST, Algorithm 1) which adjusts optimistically a candidate predictor (*e.g.*, obtained through classical OGD) with respect to an experts' advice. In what follows, we consider this experts' advice as an *additional knowledge* which consists in a sequence of vectors belonging to $\mathbb{R}^d$. In the Optimistic online learning vision, this expert's advice has to be incorporated into the algorithm of interest. Incorporating such knowledge is not a new idea and has been used for instance by Rakhlin and Sridharan (2013a;b). We choose to exploit our additional knowledge through the notion of *performance*.

**Definition 2.1.** *We use the notation $\langle x, y \rangle_H := x^T H y$ to design the inner product associated to a positive definite matrix $H$. For a sequence of additional knowledge $\nu = (\nu_t)_{t\geq 0}$, a sequence $\hat{\mu}_{temp} = (\hat{\mu}_{temp,t})_{t\geq 1} \in \mathcal{K}^{\mathbb{N}}$ (the output of a classical online procedure) and for any positive definite matrix $H$, one defines the* performance *at time $t$ of $\hat{\mu}_{temp}$ with regards to $\nu, H$ as follows: we set $m_t := \frac{\nu_{t+1} + \nu_t}{2}$ and*

$$\mathrm{Perf}(t, H, \hat{\mu}_{temp}, \nu) := \langle \hat{\mu}_{temp,t+1} - m_t, \nu_{t+1} - \nu_t \rangle_H.$$

For more information about why this notion of performance emerged, we refer to Appendix C.

**Remark 2.2.** *At time $t$, the performance exploits the additional knowledge $\nu$ through two terms: $m_t$ and $\nu_{t+1} - \nu_t$. The first term is new to the best of our knowledge while the second is similar to the experts' advice of Rakhlin and Sridharan (2013a). Indeed, the experts' advice of Rakhlin and Sridharan (2013a) is an information on the gradient space. Here, $\nu_{t+1} - \nu_t$ gives a similar information. This point is also highlighted in Jadbabaie et al. (2015) as their path $D_T$ focuses on the distance between the experts' advice and the gradient of their predictor.*

We now state the algorithm ADJUST (Algorithm 1) which takes as input $\hat{\mu}_{temp}$, $\nu$, $H, t$ as defined in definition 2.1 and outputs an updated predictor $\hat{\mu}_{t+1}$. We denote by $\Pi_{\mathcal{K},\mathcal{H}}$ the projection over the closed convex set $\mathcal{K}$ with respect to the distance induced by $\langle ., . \rangle_H$.

---

**Algorithm 1:** The ADJUST algorithm

**Parameters :** Time $t$, positive definite matrix $H$, additional knowledge $\nu$, candidate $\hat{\mu}_{temp,t+1}$

**1** Set up $m_t = \frac{\nu_{t+1} + \nu_t}{2}$.

**2** **If** $\mathrm{Perf}(t, H, \hat{\mu}_{temp}, \nu) < 0$**, then:**

**3**     Set $\hat{\mu}_{t+1} = \arg\min_{\mu \in \mathcal{K}} \|2m_t - \hat{\mu}_{temp,t+1} - \mu\|_H^2 := \Pi_{\mathcal{K},H}(2m_t - \hat{\mu}_{temp,t+1})$

**4** **Else:**

**5**     Set $\hat{\mu}_{t+1} = \arg\min_{\mu \in \mathcal{K}} \|\hat{\mu}_{temp,t+1} - \mu\|_H^2 := \Pi_{\mathcal{K},H}(\hat{\mu}_{temp,t+1})$

**6** **Return** $\hat{\mu}_{t+1}$

---

We illustrate in Fig. 1 what ADJUST concretely performs when $H = \mathbf{I}_2$ and $\mathcal{K} = \mathbb{R}^2$.

When $\mathrm{Perf}(t, I, \hat{\mu}_{temp}, \nu) < 0$, it means that $\hat{\mu}_{temp,t+1} - m_t$ does not point in the same direction than $\nu_{t+1} - \nu_t$. Thus ADJUST corrects this trajectory by taking the symmetric $2m_t - \hat{\mu}_{temp,t+1}$ of $\hat{\mu}_{temp}$: this inverts the sign of its performance. Thus, ADJUST outputs $\hat{\mu}_{t+1}$ such that the distance between $\hat{\mu}_{t+1}$ and $\nu_{t+1}$ is smaller than the one between $\hat{\mu}_{temp,t+1}$ and $\nu_{t+1}$. This is further developed in Lemma 2.3.

**Lemma 2.3.** *For all $t \geq 0$, any positive definite $H$, any $\hat{\mu}_{temp,t+1}$, $\nu_{t+1}$, $\nu_t$ defined as in ADJUST (algorithm 1): we denote by $\|.\|_H^2$ the norm associated to the scalar product $\langle ., . \rangle_H$,*

$$\|\hat{\mu}_{t+1} - \nu_{t+1}\|_H^2 \leq \|\hat{\mu}_{temp,t+1} - \nu_t\|_H^2.$$

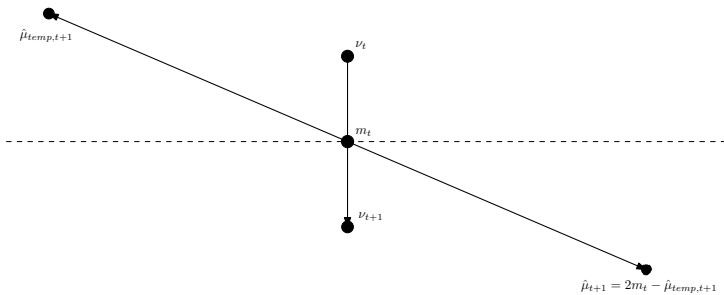

Figure 1: Action of ADJUST when performance is negative

*Proof of Lemma 2.3.* First, if $\text{Perf}(t+1, H, \hat{\mu}_{temp}, \nu) < 0$, then $\hat{\mu}_{t+1} = \Pi_{\mathcal{K},H}(2m_t - \hat{\mu}_{temp,t+1})$ and one has:

$$\|\hat{\mu}_{t+1} - \nu_{t+1}\|_H^2 = \|\Pi_{\mathcal{K},H}(2m_t - \hat{\mu}_{temp,t+1}) - \nu_{t+1}\|_H^2 \quad \leq \|2m_t - \hat{\mu}_{temp,t+1} - \nu_{t+1}\|_H^2$$
$$= \|\hat{\mu}_{temp,t+1} - \nu_t\|_H^2.$$

The last line holding thanks to the definition of $m_t$. Second, if $\text{Perf}(t, H, \hat{\mu}_{temp}, \nu) \geq 0$, we use:

**Lemma 2.4.** *We have* $\forall t \geqslant 0, \|\hat{\mu}_{temp,t+1} - \nu_{t+1}\|_H^2 = \|\hat{\mu}_{temp,t+1} - \nu_t\|_H^2 - 2\text{Perf}(t, H, \hat{\mu}_{temp}, \nu).$

*Proof of Lemma 2.4.* Recall that $m_t = \frac{\nu_{t+1}+\nu_t}{2}$. We have:

$$\|\hat{\mu}_{temp,t+1} - \nu_{t+1}\|_H^2 = \|\hat{\mu}_{temp,t+1} - m_t + m_t - \nu_{t+1}\|_H^2$$
$$= \|\hat{\mu}_{temp,t+1} - m_t\|_H^2 - \text{Perf}(t, H, \hat{\mu}_{temp}, \nu) + \frac{\|\nu_t - \nu_{t+1}\|_H^2}{4}.$$

And $\|\hat{\mu}_{temp,t+1} - \nu_t\|_H^2 = \|\hat{\mu}_{temp,t+1} - m_t\|_H^2 + \text{Perf}(t, H, \hat{\mu}_{temp}, \nu) + \frac{\|\nu_{t+1} - \nu_t\|_H^2}{4}.$

Thus,

$$\|\hat{\mu}_{temp,t+1} - \nu_{t+1}\|_H^2 = \|\hat{\mu}_{temp,t+1} - \nu_t\|_H^2 - 2\text{Perf}(t, H, \hat{\mu}_{temp}, \nu).$$

$\square$

Finally,

$$\|\hat{\mu}_{t+1} - \nu_{t+1}\|_H^2 = \|\Pi_{\mathcal{K},H}(\hat{\mu}_{temp,t+1}) - \nu_{t+1}\|_H^2 \leq \|\hat{\mu}_{temp,t+1} - \nu_{t+1}\|_H^2$$
$$= \|\hat{\mu}_{temp,t+1} - \nu_t\|_H^2 - 2\text{Perf}(t, H, \hat{\mu}_{temp}, \nu) \qquad \text{(by Lemma 2.4)}$$
$$\leq \|\hat{\mu}_{temp,t+1} - \nu_t\|_H^2.$$

The last line holding because our performance is positive in this case. This concludes the proof. $\square$

## 2.2 Choice of the additional knowledge

We propose in this section a data-driven procedure to obtain additional knowledge. We take inspiration from the OMGD algorithm (Zhang et al., 2017). We name this procedure CONSTRUCT and detail it in algorithm 2. It consists in applying $K > 0$ steps of the classical gradient descent algorithm to obtain a good aproximation of the last observed minimum.

We recall in Lemma 2.5 a convergence property of the gradient descent algorithm.

**Lemma 2.5.** *Assume the considered steps* $(\eta'_j)$ *verify for all* $j$, $\frac{1}{\eta'_j} - \lambda \leq \frac{1}{\eta'_{j-1}}$. *Then for any $t$ we have,*

---

**Algorithm 2:** The CONSTRUCT algorithm.

---

**Parameters** : The number $K$ of iterations, step-sizes $(\eta'_j)_{j=1..K}$
Current loss function $\ell_t$, current point $\hat{\mu}_t$

**Initialisation:** Set $\mathbf{x}_0 := \hat{\mu}_t$

**1 For** $j$ **in** $0..K-1$**:**

**2** Update

$$\mathbf{x}_{j+1} = \Pi_{\mathcal{K}} \left( \mathbf{x}_j - \eta'_j \nabla \ell_t(\mathbf{x}_j) \right)$$

**3 Return** $\nu_{t+1} := \frac{1}{K} \sum_{j=1}^{K} \mathbf{x}_j$

---

$$\ell_t(\nu_{t+1}) - \ell_t(\mu_t^*) \le \frac{G^2}{K} \sum_{j=1}^{K} \eta'_j.$$

Remark that it is essential to consider strongly convex functions to obtain the rate of Lemma 2.5. To satisfy the technical condition on the step sizes, we can consider the step sequence $(\frac{1}{\lambda t^\alpha})_{t \ge 1}$ for any $\alpha \in [0,1]$.

*Proof.* Let $t \ge 0$. Recall that $\nu_{t+1}$ is defined as the Polyak averaging $\nu_{t+1} := \frac{1}{K} \sum_{j=1}^{K} \mathbf{x}_j$. First, we remark that by convexity of $\ell_t$:

$$\ell_t(\nu_{t+1}) - \ell_t(\mu_t^*) = \ell_t \left( \frac{1}{K} \sum_{j=1}^{K} \mathbf{x}_j \right) - \ell_t(\mu_t^*) \le \frac{1}{K} \sum_{j=1}^{K} \ell_t(\mathbf{x}_j) - \ell_t(\mu_t^*).$$

Because CONSTRUCT is a gradient descent with steps $(\eta'_j)_{j=1..K}$ on the $\lambda$-strongly convex function $\ell_t$, one has for any $j$, the classical route of proof for static regret bound for strongly convex functions described in (Hazan, 2019, Theorem 3.3). One then has the following, which concludes the proof:

$$\sum_{j=0}^{K} (\ell_t(\mathbf{x}_j) - \ell_t(\mu_t^*)) \le G^2 \sum_{j=1}^{K} \eta'_j.$$

$\square$

## 3 Main results

**Outline.** We present in this section three variations of OGD, ONS and AdaGrad followed by theoretical garantees for D-Regret and D-C-Risk. Our theoretical result assume the CONSTRUCT algorithm but also work for any additional knowledge satisfying technical assumptions (translating here that the experts' advice at time $t$ is a good approximation of the minimum at time $t-1$).

**Proof technique.** Concerning our proof techniques, we have two strategies. On the one hand our proofs concerning the dynamic regret of our methods (resp. Thms. 3.1, 3.3 and 3.5 ) are gathered in Appendix D and consists in an adaptation of the static proofs of OGD,ONS,AdaGrad all lying in Hazan (2019). We adapt those proofs using Lemmas 2.3 and 2.5. On the other hand hand, proofs on the dynamic cumulative risk (resp. Thms. 3.2, 3.4 and 3.6) lie in Appendix E and use the same kind of argument incorporated within the SOCO framework of Wintenberger (2021) described in Appendix E.1.

### 3.1 Dynamic Online Gradient Descent

Our variation of the OGD, called Dynamic OGD (D-OGD), is presented in algorithm 6, it exploits an additional information $(\nu_t)_t$ at each time step. Its associated theoretical guarantee for D-Regret is stated in Thm. 3.1.

---

**Algorithm 3:** Dynamic Projected OGD onto a closed convex space $\mathcal{K}$.

---

**Parameters** : Epoch $T$, step-sizes $(\eta_t)$
**Initialisation:** Initial point $\mu_1 \in \mathcal{K}$, additional information $(\nu_1) \in \mathcal{K}$

**1 For** $t$ in $\{1, \ldots, T\}$:
**2**     Update $\hat{\mu}_{temp,t+1} = \hat{\mu}_t - \eta_t \nabla \ell_t(\hat{\mu}_t)$
**3**     Observe $\nu_{t+1}$,
**4** $\hat{\mu}_{t+1} = \text{ADJUST}(t, \mathbf{I}_d, \nu, \hat{\mu}_{temp,t+1})$
**5 Return** $\hat{\mu} = (\hat{\mu}_t)_{t=0..T}$

---

**Theorem 3.1.** *Denote by $\mu_t^* = \operatorname{argmin}_{\mu \in \mathcal{K}} \ell_t(\mu)$. We assume that our predictors $\hat{\mu}$ are obtained using D-OGD (Algorithm 6) with steps $\eta = (\frac{D}{G\sqrt{t}})_{t=1..T}$. We also assume our additional knowledge $\nu$ to be the output of CONSTRUCT (Algorithm 2) used at time $t$ with steps $\eta' = (\frac{1}{\lambda j})_{j=1..K}$ and $K = \lceil \sqrt{T} \rceil$. Then, dynamic regret of D-OGD with regards to $\mu^* = (\mu_t^*)_{t=0..T}$ the true minimisers satisfy :*

$$\sum_{t=1}^{T} \ell_t(\hat{\mu}_t) - \sum_{t=1}^{T} \ell_t(\mu_t^*) \le GP_T(\nu) - \lambda S_T(\nu) + \frac{3}{2}GD\sqrt{T} + \frac{G^2}{\lambda}(1 + \log(1+T))\sqrt{T}.$$

*Furthermore, this result remains for any additional knowlege $\nu$ such that for any $t$, $\ell_t(\nu_{t+1}) - \ell_t(\mu_t^*) = \mathcal{O}(\log(t)/\sqrt{t})$.*

Thm. 3.1 provides a worst-case guarantee for the dynamic regret of D-OGD. An interesting point is that our bound decoupled the influence of the paths lengths from the horizon $T$, which is not usual in the literature (Zinkevich, 2003 proposed a bound of $\mathcal{O}(\sqrt{T}(1 + P_T))$ later improved in Zhang et al. (2018a) in a $\mathcal{O}(\sqrt{T(1 + P_T)})$).

Note that here $K = \lceil \sqrt{T} \rceil$ is determined with respect to a fixed horizon $T$. When we do not know in advance the stopping time of D-OGD, we can apply CONSTRUCT at each time $t$ with the evolutive number of iterations $K_t = \lceil \sqrt{t} \rceil$. This leads to a D-Regret bound with the same order of magnitude. We did not detail this point in Thm. 3.1 for the sake of clarity.

The price to pay for a good additional knowledge is the time dedicated to CONSTRUCT ($\lceil \sqrt{T} \rceil$ iterations at each time step). This is not new as a similar time complexity arose in the OMGD algorithm of Zhang et al. (2017), when one chooses the step-size $\eta = 1/\sqrt{T}$. Furthermore, if we consider that the true minimiser $\mu_t^*$ is revealed to the learner at time $t+1$, then we can use it as experts advice and the guarantee of Thm. 3.1 still holds. Thus, $P_T(\nu) = P_{T-1}(\mu^*) + ||\mu_1^* - \nu_1||$. This allow us to compare in this case, our results with those of Zhang et al. (2017). We then notice that our convergence rate is worse than their $\mathcal{O}(\min(P_T(\mu^*), S_T(\mu^*)))$. However, we only assumed our function to be strongly convex (while Zhang et al. (2017) added a smoothness assumption on the losses) and our result allow the use of experts to maintain a fully empirical upperbound even when the true minimisers are hidden to the learner.

**Theorem 3.2.** *We assume that our predictors $\hat{\mu}$ are obtained using D-OGD(Algorithm 6) with steps $\eta = (\frac{D}{G\sqrt{t}})_{t=1..T}$. We also assume our additional knowledge $\nu$ to be the output of CONSTRUCT (Algorithm 2) used at time $t$ with steps $\eta' = (\frac{1}{\lambda j})_{j=1..K}$ and $K = \lceil \sqrt{T} \rceil$. Then, dynamic cumulative risk satisfies with probability $1 - 3\delta$, for any $T \ge 1$, for any sequence $(\mu_t)_{t=1..T}$ such that $\mu_t$ is $\mathcal{F}_{t-1}$-measurable:*

$$\sum_{t=1}^{T} L_t(\hat{\mu}_t) - \sum_{t=1}^{T} L_t(\mu_t) \le GP_T(\nu) - \lambda S_T(\nu) + \tilde{\mathcal{O}}(\sqrt{T})$$

*where the $\tilde{\mathcal{O}}$ hides a log factor. Furthermore, this result remains for any additional knowlege $\nu$ such that for any $t$, $\ell_t(\nu_{t+1}) - \ell_t(\mu_t^*) = \mathcal{O}(\log(t)/\sqrt{t})$.*

First of all, Thm. 3.2 hold for any predictable sequence of comparators $\mu$ which are not involved on the upper bound. This choice has been made to maintain a fully empirical upper bound: indeed, predictable

sequences are often unknown as they may depend on the conditional distribution of the data (unknown in practice). This bound ensures us that D-OGD allow us to nearly maintain the same convergence rate than Thm. 3.1 when the controlled quantity is now a conditioned risk $\mathbb{E}_{t-1}[\ell]$ instead of an empirical loss $\ell$. This shift alllows our guarantees to ensure a good generalisation ability of our predictors. An interesting point is that our upper bound is empirical while the left hand side is theoretical: we have a computable guarantee about how robust are our predictors to the intrinsic randomness of the considered problem. Note that our result holds for any sequence $\mu$ such that $\mu_t$ is $\mathcal{F}_{t-1}$-measurable. We present in Sec. 4.2 a toy experiment which exploits this additional flexibility by showing it may not be relevant to compare ourselves to the true minimisers $\mu^*$.

### 3.2 Dynamic Online Newton Step

Algorithm 4 details the D-ONS algorithm, which is an updated version of the ONS (Hazan et al., 2007) and we present in Thm. 3.3 its associated D-Regret bound.

---

**Algorithm 4:** Dynamic ONS onto a closed convex space $\mathcal{K}$.

**Parameters** : Epoch $T$, step $\gamma, \varepsilon > 0$.
**Initialisation:** convex set $\mathcal{K}$, initial point $\mu_1 \in \mathcal{K} \subseteq \mathbb{R}^d$, additional information $\nu_1 \in \mathcal{K}, A_0 = \varepsilon I_d$

1 **For** $t$ in $\{1, \ldots, T\}$:
2      Update $A_t = A_{t-1} + \nabla_t \nabla_t^\top$
3      Set $\hat{\mu}_{temp,t+1} = \hat{\mu}_t - \frac{1}{\gamma} A_t^{-1} \nabla_t$
4      Observe $\nu_{t+1}$
5      $\hat{\mu}_{t+1} = \text{ADJUST}(t, A_t, \nu, \hat{\mu}_{temp,t+1})$
6 **Return** $\hat{\mu} = (\hat{\mu}_t)_{t=0..T}$

---

**Theorem 3.3.** *Denote by $\mu_t^* = \text{argmin}_{\mu \in \mathcal{K}} \ell_t(\mu)$. We assume that our predictors $\hat{\mu}$ are obtained using D-ONS (Algorithm 4 ) with $\gamma = \frac{1}{2}\min\left\{\frac{1}{GD}, \alpha\right\}$, $\varepsilon = \frac{1}{\gamma^2 D^2}$. We also assume our additional knowledge $\nu$ to be the output of CONSTRUCT (Algorithm 2) used at time $t$ with steps $\eta' = (\frac{1}{\lambda j})_{j=1..K}$ and $K = T$. Then, dynamic regret of D-ONS with regards to $\mu^* = (\mu_t^*)_{t=0..T}$ the true minimisers satisfy :*

$$\sum_{t=1}^{T} \ell_t(\hat{\mu}_t) - \sum_{t=1}^{T} \ell_t(\mu_t^*) \leq GP_T(\nu) - \lambda S_T(\nu) + 2\left(\frac{G^2}{\lambda}(d+1) + dGD\right)(1 + \log(T)).$$

*Furthermore, this result remains for any additional knowlege $\nu$ such that for any $t$, $\ell_t(\nu_{t+1}) - \ell_t(\mu_t^*) = \mathcal{O}(1/t)$.*

Thm. 3.3 can be compared to the Online Multiple Newton Update (OMNU) of Zhang et al. (2017) which proposed a competitive rate of $\mathcal{O}(\min(P_T, S_T))$ for OMNU. While our rate is weaker than theirs, our results hold with the single assumption of strong convexity. Indeed, Zhang et al. (2017, Thm 11.) holds under a set of technical assumptions Zhang et al. (2017, Assumption 10) involving among others, the strict convexity of the losses and holding for problems having small variations of their successive minimas. Our result requires less assumptions but comes at the cost of $K = T$ iterations of ADJUST at each time step. Finally, taking $K_t = t$ at each time step allow us to not knowing in advance the stopping time of D-ONS and recovers a slighlty deteriorated rate of $\mathcal{O}(d\log(T)^2)$.

**Theorem 3.4.** *We assume that our predictors $\hat{\mu}$ are obtained using D-ONS(Algorithm 4) with $\gamma = \frac{1}{2}\min\left\{\frac{1}{GD}, \frac{\alpha}{4}\right\}$, $\varepsilon = \frac{1}{\gamma^2 D^2}$. We also assume our additional knowledge $\nu$ to be the output of CONSTRUCT (Algorithm 2) used at time $t$ with steps $\eta' = (\frac{1}{\lambda j})_{j=1..K}$ and $K = T$. Then, the dynamic cumulative risk satisfies with probability $1 - 2\delta$, for any $T \geq 1$, for any sequence $(\mu_t)_{t=1..T}$ such that $\mu_t$ is $\mathcal{F}_{t-1}$-measurable:*

$$\sum_{t=1}^{T} L_t(\hat{\mu}_t) - \sum_{t=1}^{T} L_t(\mu_t) \leq GP_T(\nu) + 2G^2 S_T(\nu) + \mathcal{O}(d\log(T) + \log(1/\delta)),$$

*where $L_t = \mathbb{E}_{t-1}[\ell_t]$. Furthermore, this result remains for any additional knowlege $\nu$ such that for any $t$, $\ell_t(\nu_{t+1}) - \ell_t(\mu_t^*) = \mathcal{O}(1/t)$.*

### 3.3 Dynamic AdaGrad

Algorithm 5 details the D-AdaGrad algorithm, which is an updated version of AdaGrad (Duchi et al., 2011) and we present in Thm. 3.5 its associated D-Regret bound. We use the notation $A \bullet B$ to denote the element-wise multipication between the matrices $A$ and $B$.

---

**Algorithm 5:** Dynamic AdaGrad onto a closed convex space $\mathcal{K}$.

**Parameters** : Epoch T, step $\eta$, parameter $\varepsilon$.

**Initialisation:** Initial point $\mu_1 \in \mathcal{K}$, additional information $(\nu_1) \in \mathcal{K}$, $G_0 = \varepsilon \mathbf{I}_d, H_0 = G_0^{1/2}$

**1 For** $t$ in $\{1, \dots, T\}$:

**2**      Update $G_t = G_{t-1} + \nabla_t \nabla_t^\top$

**3**      Update $H_t = \underset{H \succeq 0}{\arg\min} \left\{ G_t \bullet H^{-1} + \mathrm{Tr}(H) \right\} = G_t^{1/2}$

**4**      Set $\hat{\mu}_{temp,t+1} = \hat{\mu}_t - \eta H_t^{-1} \nabla_t$

**5**      Observe $\nu_{t+1}$

**6**      $\hat{\mu}_{t+1} = \text{ADJUST}(t, H_t, \nu, \hat{\mu}_{temp,t+1})$

**7 Return** $\hat{\mu} = (\hat{\mu}_t)_{t=0..T}$

---

**Theorem 3.5.** *Denote by $\mu_t^* = \mathrm{argmin}_{\mu \in \mathcal{K}} \ell_t(\mu)$. We assume that our predictors $\hat{\mu}$ are obtained using D-AdaGrad (Algorithm 5 ) with with $\varepsilon = \frac{2}{D^2}, \eta = \frac{D}{\sqrt{2}}$. We also assume our additional knowledge $\nu$ to be the output of CONSTRUCT (Algorithm 2) used at time $t$ with steps $\eta' = (\frac{1}{\lambda j})_{j=1..K}$ and $K = T$. Then, dynamic regret of D-AdaGrad with regards to $\mu^* = (\mu_t^*)_{t=0..T}$ the true minimisers satisfy :*

$$\sum_{t=1}^{T} \ell_t(\hat{\mu}_t) - \sum_{t=1}^{T} \ell_t(\mu_t^*) \leq GP_T(\nu) - \lambda S_T(\nu) + \sqrt{2}D \left( 1 + \sqrt{\min_{H \in \mathcal{H}} \sum_t \|\nabla_t\|_H^{*2}} \right) + \frac{G^2}{\lambda}(1 + \log(T)).$$

*Furthermore, this result remains for any additional knowlege $\nu$ such that for any $t$, $\ell_t(\nu_{t+1}) - \ell_t(\mu_t^*) = \mathcal{O}(1/t)$.*

Thm. 3.5 nearly recovers the convergence rate of AdaGrad for static regret at the cost of an extra path length and $\mathcal{O}(\log(T))$ factor. Note that, as in Thm. 3.3, the evolutive iteration number $K_t = t$ can be chosen instead of $K = T$ to make the procedure valid for any horizon $T$ (not necessarily fixed in advance) at the cost of an extra log factor.

Furthermore, Thm. 3.5 goes beyond the scope of Zhang et al. (2017), as methods proposed in this paper do not deal with AdaGrad. Note that our approach is not the first to propose a dynamic regret bound for AdaGrad (see the recent work of Nazari and Khorram, 2022) but we point that our approach is general enough to provide bounds simultaneaously for variants of OGD, ONS, AdaGrad. However our approach is, to our knowledge, the first whih also proposes dynamic guarantees for the cumulative risk (*i.e.,* regret with losses $\mathbb{E}_{t-1}[\ell_t]$) as stated below.

**Theorem 3.6.** *We assume that our predictors $\hat{\mu}$ are obtained using D-AdaGrad (Algorithm 5 ) with with $\varepsilon = \frac{2}{D^2}, \eta = \frac{D}{\sqrt{2}}$. We also assume our additional knowledge $\nu$ to be the output of CONSTRUCT (Algorithm 2) used at time $t$ with steps $\eta' = (\frac{1}{\lambda j})_{j=1..K}$ and $K = T$. Then, dynamic cumulative risk satisfies with probability $1 - 2\delta$, for any $T \geq 1$, for any sequence $(\mu_t)_{t=1..T}$ such that $\mu_t$ is $\mathcal{F}_{t-1}$-measurable:*

$$\sum_{t=1}^{T} L_t(\hat{\mu}_t) - \sum_{t=1}^{T} L_t(\mu_t) \leq GP_T(\nu) + \mathcal{O} \left( \sqrt{\min_{H \in \mathcal{H}} \sum_t \|\nabla_t\|_H^{*2}} + \log \frac{T}{\delta} \right).$$

*Note that this result stll holds for any additional knowlege $\nu$ such that for any $t$, $\ell_t(\nu_{t+1}) - \ell_t(\mu_t^*) = \mathcal{O}(1/t)$.*

# 4 Experiments

**Experiments.** We propose two sets of experiments. The first one gathers 4 classical dataset two regression and two classification problems. Its goal is to assess our algorithm's efficiency by plotting the averaged cumulative losses $\sum_{i=1}^{t} \ell(h_i, z_i)/t$ at any time $t$. The second experiment is a toy example designed to show that D-C-Risk is a relevant tool to handle learning processes on noisy problems. For those two experiments we compute three algorithms: the celebrated Online Gradient Descent (Zinkevich, 2003, Alg. 1), the D-OGD algorithm (Algorithm 6) and a variant of the Online Multiple Gradient Descent (OMGD) algorithm with decreasing steps (Zhang et al., 2017, Alg. 1).

The reason we computed OMGD is that CONSTRUCT (Algorithm 2) is following the same idea as OMGD (*i.e.*, performing a gradient descent at each time step for more accurate predictors). An interesting question is whether D-OGD provides similar or better results than OMGD? We address this below. Furthermore, we would expect that using the output of CONSTRUCT as additional knowledge instead of predictor would provide us an additional flexibility in our learning process, is it the case in practice?

## 4.1 Experiments on real-life datasets

We conduct experiments on a few real-life datasets, in classification and regression. Our objective is twofold: check the convergence of our learning methods and compare their efficiencies with classical algorithms.

**Binary Classification.** At each round $t$ the learner receives a data point $x_t \in \mathbb{R}^d$ and predicts its label $y_t \in \{-1, +1\}$ using $\langle x_t, h_t \rangle$, with $h_t$ being the predictor given by the online algorithm of interest. The adversary reveals the true value $y_t$, then the learner suffers the loss $\ell(h_t, z_t) = \left(1 - y_t h_t^T x_t\right)_+$ with $z_t = (x_t, y_t)$ and $a_+ = a$ if $a > 0$ and $a_+ = 0$ otherwise.

**Linear Regression.** At each round $t$, the learner receives a set of features $x_t \in \mathbb{R}^d$ and predicts $y_t \in \mathbb{R}$ using $\langle x_t, h_t \rangle$ with $h_t$ being the predictor given by the online algorithm of interest. Then the adversary reveals the true value $y_t$ and the learner suffers the loss $\ell(h_t, z_t) = \left(y_t - h_t^T x_t\right)^2$ with $z_t = (x_t, y_t)$.

**Datasets.** We consider four real world dataset: two for classification (Breast Cancer and Pima Indians), and two for regression (Boston Housing and California Housing). All datasets except the Pima Indians have been directly extracted from `sklearn` (Pedregosa et al., 2011). Breast Cancer dataset (Street et al., 1993) is available here and comes from the UCI ML repository as well as the Boston Housing dataset (Belsley et al., 2005) which can be obtained here. California Housing dataset (Pace and Barry, 1997) comes from the StatLib repository and is available here. Finally, Pima Indians dataset (Smith et al., 1988) has been recovered from this Kaggle repository. Note that we randomly permuted the observations to avoid to learn irrelevant human ordering of data (such that date or label).

**Parameter settings.** We ran our experiments on a 2021 MacBookPro with an M1 chip and 16 Gb RAM. For OGD, the initialisation point is $\mathbf{0}_{\mathbb{R}^d}$ and the values of the learning rates are set to $\eta = 1/2\sqrt{m}$. where $m$ is the size of the considered dataset. For OMGD, we ran the procedure while, at time $t$, performing a gradient descent with $K = 100$ iterations. This auxiliary gradient descent has been performed with steps $(\lambda/2\sqrt{j})_{j=1..K}$. $\lambda$ ,being an empirical stabiliser set to $0.1/\sqrt{m}$. For D-OGD, we ran the procedure with a constant step $\eta = 0.1/\sqrt{m}$. We ran CONSTRUCT to generate our additional knowledge with the iteration number $K = 100$ and steps $(\eta'_j)_{j=1..K} = (\lambda/2\sqrt{j})_{j=1..K}$, $\lambda$ ,being an empirical stabiliser set to $0.1/\sqrt{m}$.

**Quantity of interest.** For each dataset, we plot the evolution of the averaged cumulative loss $\sum_{i=1}^{t} \ell(h_i, z_i)/t$ as a function of the step $t = 1, ..., m$, where $m$ is the dataset size and $h_i$ is the decision made by the learner $h_i$ at step $i$. The results are gathered in Fig. 2.

**Empirical findings.** On those datasets, OMGD with adaptive steps and D-OGD seems to perform rather equivalently, except on the PIMA Indians dataset where D-OGD outperforms OMGD. On two datasets (Breast Cancer and California Housing), D-OGD performs better than OGD, otherwise both methods performs similarly. A reason that could explain the efficiency of our method compared to OMGD in the PIMA Indians dataset is that because this problem is difficult (*i.e.,* noisy), the technical condition stated in (Zhang

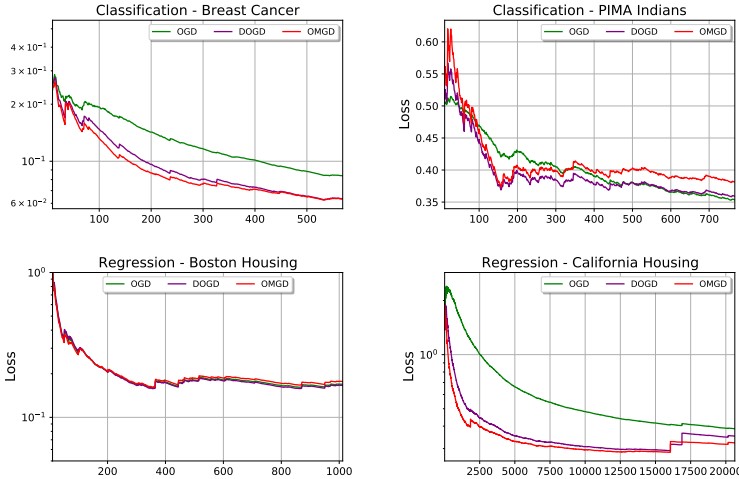

Figure 2: Averaged cumulative losses for all four considered datasets. The $x$-axis is the time.

et al., 2017, Corollary 4) may not be statisfied. This would impeach OMGD to attain competitive results. Furthermore, note that in any case, D-OGD is at least as good as OGD or OMGD.

### 4.2 A toy experiment: the Online Quadratic Problem

**Theoretical framework.** Our problem is set as follows: at each time step $t$, a random variable $\theta_t$ is drawn. For all $t$, $\theta_t$ is such that

$$P_t = \mathcal{L}(\theta_t \mid \mathcal{F}_{t-1}) = \mathcal{N}(\mathtt{moy}_t, \sigma_t^2).$$

We assume that there exists $D_m, D_\sigma$ positive values such that for all $t$, $(\mathtt{moy}_t, \sigma_t) \in [-D_m/2, D_m/2] \times [0; D_\sigma]$. Finally, we consider at time $t$, the loss $\ell_t(\theta) = (\theta_t - \theta)^2$. We refer to this framework as the *Online Quadratic Problem*.

**Quantity of interest.** We study the D-C-Risk w.r.t. the sequence $\mu_t = \mathtt{moy}_t$. We cannot compare ourselves to the true minimiser $\mu_t^* = \theta_t$ because this quantity is not $\mathcal{F}_{t-1}$ measurable. However, we show below that there exists another meaningful comparator. Indeed, in our setup, we precised that $\mathtt{moy}_t$ was assumed to be $\mathcal{F}_{t-1}$-measurable so let us see what gives the dynamic cumulative risk for any sequence of predictors $(\hat{\mu}_t)_{t\geq 0}$:

$$\sum_{t=1}^{T} L_t(\hat{\mu}_t) - \sum_{t=1}^{T} L_t(\mathtt{moy}_t) = \sum_{t=1}^{T} \mathbb{E}_{t-1}[(\theta_t - \hat{\mu}_t)^2] - \sum_{t=1}^{T} \underbrace{\mathbb{E}_{t-1}[(\theta_t - \mathtt{moy}_t)^2]}_{=\sigma_t^2} = \sum_{t=1}^{T} (\hat{\mu}_t - \mathtt{moy}_t)^2.$$

The last line holding thanks to a bias-variance tradeoff. This basic calculation shows that for this learning problem, using $(\mathtt{moy}_t)_t$ as comparators instead of the true minimisers leads to a meaningful regret. Yet, we can derive from the general notion of dynamic regret a comparison between our prediction and the true mean of the data. One will see in the experiments that D-OGD can approximate the means better than classical OGD at high times.

**Parameter settings.** All our algorithms are using a projection on the ball centered in 0 of diameter $D = 10$. For OGD, the initialisation point is $\mathbf{0}_{\mathbb{R}^d}$ and the values of the learning rates are set to $\eta_t = 1/2\sqrt{t}$. For OMGD, we ran the procedure while, at time $t$, performing a gradient descent with $K = 100$ iterations. This auxiliary gradient descent has been performed at time $t$ with steps $(\lambda_t/2\sqrt{j})_{j=1..K}$, $\lambda_t$ being an empirical stabiliser set to $1/2\sqrt{t}$. For D-OGD, we ran two variants: the first uses CONSTRUCT to generate our additional knowledge. We run algorithm 6 with steps $\eta_t = 1/2\sqrt{t}$ at time $t$. We run CONSTRUCT with, at each time $t$, the iteration number $K = 100$ and steps $(\eta'_j)_{j=1..K} = (\lambda_t/2\sqrt{j})_{j=1..K}$, $\lambda_t$ being an empirical stabiliser set

to $1/2\sqrt{t}$. The second does not use CONSTRUCT and instead defines at each time $t$ $\nu_{t+1} \sim \mathcal{N}(\hat{\mu}_t, \sigma_1^2)$ with $\sigma_1 = 0.4$. Similarly, we run algorithm 6 with steps $\eta_t = 1/2\sqrt{t}$ at time $t$.

**Experimental framework.** We take for any $t$, $\texttt{moy}_t = \sin\left(\frac{t}{\omega}\right)$ with $\omega = 200$, yet the means are a deterministic sequence fixed before our study. Then our $\theta_t$ are drawn independently. We also fix for any $t$, $\sigma_t = \sigma = 4$. We chose $K$ (the number of iterations to acquire our additional knowledge) equal to 100. Results are gathered in Fig. 3.

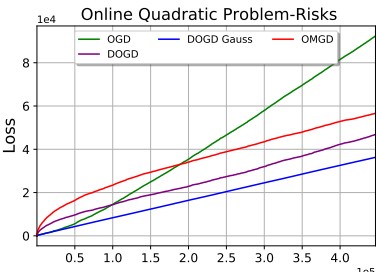 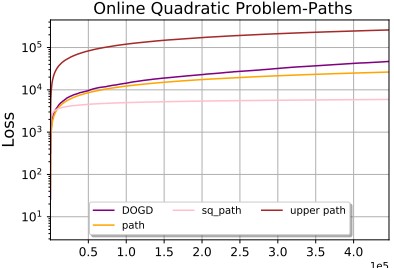

Figure 3: On the left, cumulative risks of D-OGD (purple,blue), OMGD (red), OGD (green). On the right, plot of D-OGD and its associated paths. The $x$-axis s the time. $\texttt{sq\_path}$ is $S_t(\nu)$, $\texttt{path}$ is $P_t(\nu)$, $\texttt{upper\_path}$ is $GP_t(\nu) - \lambda S_t(\nu)$.

**Empirical findings.** First, OGD fails on this example as the problem is too noisy: OGD fails to detect any statistical pattern between the successive points. Second, OMGD performs better than OGD but is significantly worse than D-OGD (the difference of the dynamic cumulative risks is of magnitude $10^4$). This shows that our method, which only uses the output of the auxiliary gradient descent as addtional knowledge (and not as predictors as in OMGD) provides an additional flexibility which translates onto a greater performance for extremely noisy problems. A reason that could explain the efficiency of our method compared to OMGD is again that the intrinsic noise is so high that the technical condition stated in (Zhang et al., 2017, Cor. 4) may not be statisfied, which impeachs OMGD to attain a competitive dynamic regret in $\mathcal{O}(\min(P_T^*, S_T^*))$. Finally, note that interestingly, our variant of D-OGD (the curve 'D-OGD Gauss' which uses an alternative source of additional information) provides better results here while we have no theoretical guarantee of its efficiency. This opens the way to a broader reflexion to the choice of the additional knowledge within D-OGD.

## 5 Conclusion

We provided dynamic adaptations of classical online methods. Those adaptations involved optimism through the ADJUST algorithm. We required our additional knowledge at time $i + 1$ to be a good approximation of the minimium at time $i$ to obtain good regret guarantees. A novelty of our approach is to propose a way (the CONSTRUCT algorithm) to craft this addtional information, providing a ready-to-use version of our algorithms. However, even though CONSTRUCT appeared naturally in our study as we considered OMGD of Zhang et al. (2017), it is not the only possible choice. For instance, we could use the classical Newton algorithm instead. This may be a more suited choice when the problem dimension is not too big as Newton methods are known to converge quickly (at the cost of the calculus of an inverse matrix at each time step): this would be faster than CONSTRUCT then. This instance exhibit an experimental tradeoff between accuracy and time complexity involving the dimension as hyperparameter of the problem.

However, our discussions about the choice of additional knowledge $\nu$ are driven by a narrowed vision of our additionnal knowledge: in this work, it only focuses on being a good approximation of the minimas while our bounds suggest us a deeper vision. Indeed, our bounds involve a broader tradeoff on $\nu$: on the one's hand, we make appear path lengths which invite us to consider that the sequence $\nu$ is not evolving too fast (*i.e.,* only small shifts from the additionnal knowledge through time are recommended) and on the other hand, we still require $\nu$ to be a good approximation of the past minimisers. Finding a $\nu$ optimising this tradeoff appears to be a promising route of study for future works.

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

# A    Technical background

## A.1    Azuma-Hoeffding's inequality

One recalls the celebrated Azuma- Hoeffding inequality

**Proposition A.1.** *Let $\{X_0, X_1, \cdots\}$ be a martingale with respect to filtration $\{\mathcal{F}_0, \mathcal{F}_1, \cdots\}$. Assume there are predictable processes $\{A_0, A_1, \cdots\}$ and $\{B_0, B_1, \ldots\}$ with respect to $\{\mathcal{F}_0, \mathcal{F}_1, \cdots\}$, i.e. for all $t$, $A_t, B_t$ are $\mathcal{F}_{t-1}$-measurable, and constants $0 < c_1, c_2, \cdots < \infty$ such that*

$$A_t \leq X_t - X_{t-1} \leq B_t \quad and \quad B_t - A_t \leq c_t$$

*almost surely. Then for all $\epsilon > 0$,*

$$\mathrm{P}\left(|X_n - X_0| \geq \epsilon\right) \leq 2\exp\left(-\frac{2\epsilon^2}{\sum_{t=1}^n c_t^2}\right)$$

In this work we use Azuma-Hoeffding's bound in the particular case where $A_t, B_t$ are constants almost surely.

# B  About the interest of Adjust

We discuss here about the interest of choosing ADJUST as learning algorithms. First of all, we introduce a framework which describe both OGD, OMGD and D-OGD as optimistic online algorithms.

**An optimistic class of algorithms including OGD, OMGD and D-OGD.**  In the optimistic framework, we are given experts advices materialised as addtional knowledge $\nu$ being a sequence of vectors in $\mathbb{R}^d$. We then define an *optimistic gradient-based (OGB) algorithm with judge $f$* as an online algorithm satisfying the following pattern:

---
**Algorithm 6:** An OGB algorithm with judge $f$.

---
**Parameters** : Epoch $T$, step-sizes $(\eta_t)$
**Initialisation:** Initial point $\mu_1 \in \mathcal{K}$, additional information $(\nu_1) \in \mathcal{K}$
1  **For** $t$ in $\{1, \ldots, T\}$:
2      Update $\hat{\mu}_{temp,t+1} = \hat{\mu}_t - \eta_t \nabla \ell_t(\hat{\mu}_t)$
3      Observe $\nu_{t+1}$,
4      $\hat{\mu}_{t+1} = f(t, \nu, \hat{\mu}_{temp,t+1})$
5  **Return** $\hat{\mu} = (\hat{\mu}_t)_{t=0..T}$

---

The role of the judge $f$ within an OGB algorithm is to determine, at time $t$, how we combine the additional knowledge $\nu$ with $\hat{\mu}_{temp,t+1}$ being the output of a gradient descent step. The choice of the judge depends on the confidence we have on the quality of the additional knowledge $\nu$.

We now assume here that our additional knowledge is given by CONSTRUCT. In this case both OMGD and OGD are optimistic gradient-based. Indeed:

- OMGD is an OGB algorithm with judge $f(t, \nu, \hat{\mu}_{temp,t+1}) = \nu_{t+1}$. This corresponds to the case where the judge estimate that the additional knowledge is perfectly relevant for the next prediction.

- OGD is an OGB algorithm with judge $f(t, \nu, \hat{\mu}_{temp,t+1}) = \hat{\mu}_{temp,t+1}$. This corresponds to the case where the judge estimates that the additional knowledge is useless or adveresarial, it will then choose to ignore it.

- Between those two extremes lies D-OGD which is an OGB algorithm with judge $f = $ ADJUST. This corresponds to the case of a moderate judge which tries to find a balance between the impact of the gradient descent step (an exploration phase) and the one of the additional knowledge (an exploitation one)

Thus, both OMGD and OGD can be interpreted as Optimistic algorithms treating naively the additional knowledge. OMGD leads to tighter bound than D-OGD but implies a total confidence on $\nu$: this may be suboptimal in practice as shown in Sec. 4.2.

Conversely, not exploiting at all additional knowledge leads to OGD (which is also equivalent to perform D-OGD with constant additional knowledge). We then recover dynamic regret bounds of OGD but we did not exploit the optimistic framework here.

**D-OGD as a flexible OGB algorithm with dynamic regret bounds.**  As noticed above, it is possible to recover OMGD and OGD as OBD algorithm whose judges choose either to fully trust or ignore the additional knowledge. Note that those judges do not truly combine additional knowledge and a gradient descent. However, optimistic online learning aims to combine those quantities: this is for instance what proposes Rakhlin and Sridharan (2013a)s Optimistic Mirror Descent which has static regret bounds.

A legitimate question is then: is it possible to design an optimistic online algorithm which perpetrates the way Optimistic Mirror Descent deals with additional knowledge while achieving dynamic regret bounds?

The underlying research field behind this question is not new as it has been already investigated, for instance, by Jadbabaie et al. (2015) They succeeded to provide an adaptive version of OMD (at the cost of a Lipschitz assumption for the Bregman divergence) when the additional knowledge is given on the gradient space.

We contribute to this line of work through three optimistic algorithms based on ADJUST all valid for strongly convex functions. Our algorithms hold when our additional knowledge $\nu$ is taken over the predictor space (and not the gradient one), which is less restrictive as we can still obtain a surrogate information about the gradient through $\nu_{t+1} - \nu_t$. We succeed to maintain theoretical guarantees for our elaborated optimistic strategy, but this comes at the cost of a deteriorated convergence rate with respect to the naive optimistic strategy underlying OMGD.

We insist on the fact that the interest of ADJUST is not to tighten existing bounds. Instead, we provide algorithms with non-naive optimistic strategies holding for an additional knowledge lying on the predictor space. We also furnish theoretical guarantees. Furthermore, note that D-OGD is valid for any additional knowledge satisfying technical conditions: this goes beyond the range of OMGD.

We also provide an experimental setup where D-OGD performs better than OGD and OMGD: the Online Quadratic Problem of Sec. 4.2. This situation exhibits a noisy problem, where trusting totally our experts (OMGD) or ignoring them totally (OGD) leads to poorer cumulative risks than D-OGD. This illustrates the relevance of our method to deal efficiently with additional knowledge.

## C   Inspiration for our notion of performance

Let $\eta = (\eta_t)_{t=1..T)}$ be a positive step sequence.

We denote by $\hat{\mu}_t, t \geq 1$ the sequence of predictors defined by the classical projected OGD:

$$\hat{\mu}_{t+1} = \Pi_{\mathcal{K}} \left( \hat{\mu}_t - \nabla \ell_t(\hat{\mu}_t) \right)$$

**Theorem C.1.** *Dynamic regret of projected OGD on a closed convex $\mathcal{K}$ for convex losses with steps $\eta = (\eta_t)_{t=1..T)}$ with regards to $\mu = (\mu_t)_{t=0..T} \in \mathcal{K}^T$ satisfies :*

$$\sum_{t=1}^{T} \ell_t(\hat{\mu}_t) - \sum_{t=1}^{T} \ell_t(\mu_t) \leq \sum_{t=1}^{T} \langle \nabla(\ell_t), \hat{\mu}_t - \mu_t \rangle$$

$$\leq \frac{D^2}{2\eta_T} + \frac{G^2}{2} \sum_{t=1}^{T} \eta_t - \sum_{t=1}^{T} \frac{\text{Perf}(t, \hat{\mu}, \mu)}{\eta_t}.$$

*Proof.* First, convexity of the losses gives us :

$$\sum_{t=1}^{T} \ell_t \left( \hat{\mu}_t \right) - \sum_{t=1}^{T} \ell_t \left( \mu_t \right) \leqslant \sum_{t=1}^{T} \langle \nabla \ell_t \left( \hat{\mu}_t \right), \hat{\mu}_t - \mu_t \rangle$$

To control the right hand side of this bound we use:

$$\|\hat{\mu}_{t+1} - \mu_t\|^2 \leqslant \|\hat{\mu}_t - \eta_t \nabla \ell_t \left( \hat{\mu}_t \right) - \mu_t\|^2$$

$$= \|\hat{\mu}_t - \mu_t\|^2 - 2\eta_t \langle \nabla \ell_t \left( \hat{\mu}_t \right), \hat{\mu}_t - \mu_t \rangle + \eta_t^2 \|\nabla \ell_t \left( \hat{\mu}_t \right)\|^2$$

Hence:

$$\|\hat{\mu}_{t+1} - \mu_{t+1}\|^2 \leqslant \|\hat{\mu}_t - \mu_t\|^2 - 2\eta_t \langle \nabla \ell_t \left( \hat{\mu}_t \right), \hat{\mu}_t - \mu_t \rangle + \eta_t^2 G^2 - 2\text{Perf} \left( t, \hat{\mu}, \mu \right)$$

So:

$$\langle \nabla \ell_t \left( \hat{\mu}_t \right), \hat{\mu}_t - \mu_t \rangle \leqslant \frac{\|\hat{\mu}_t - \mu_t\|^2 - \|\hat{\mu}_{t+1} - \mu_{t+1}\|^2}{2\eta_t} + \frac{\eta_t G^2}{2} - \frac{\text{Perf}(t, \hat{\mu}, \mu)}{\eta_t}$$

Summing on $t$ gives (assuming $1/\eta_0 = 0$):

$$\sum_{t=1}^T \langle \nabla \ell_t \left( \hat{\mu}_t \right), \hat{\mu}_t - \mu_t \rangle \leq \sum_{t=1}^T \|\hat{\mu}_t - \mu_t\|^2 \left( \frac{1}{2\eta_t} - \frac{1}{2\eta_{t-1}} \right) + \frac{G^2}{2} \sum_{t=1}^T \eta_t - \sum_{t=1}^T \frac{\text{Perf}(t, \hat{\mu}, \mu)}{\eta_t}$$

$$\leq D^2 \sum_{t=1}^T \left( \frac{1}{2\eta_t} - \frac{1}{2\eta_{t-1}} \right) + \frac{G^2}{2} \sum_{t=1}^T \eta_t - \sum_{t=1}^T \frac{\text{Perf}(t, \hat{\mu}, \mu)}{\eta_t}$$

$$\leq \frac{D^2}{\eta_T} + \frac{G^2}{2} \sum_{t=1}^T \eta_t - \sum_{t=1}^T \frac{\text{Perf}(t, \hat{\mu}, \mu)}{\eta_t}$$

$\square$

One can also have a stronger result for $\lambda$-strongly convex functions with the following additional assumption: We assume that our steps $\eta_t$ are such that:

$$\frac{1}{\eta_t} - \lambda \leq \frac{1}{\eta_{t-1}}$$

**Theorem C.2.** *Dynamic regret of projected OGD on a closed convex $\mathcal{K}$ with steps $\eta = (\eta_t)_{t=1..T}$ with regards to $\mu = (\mu_t)_{t=0..T} \in \mathcal{K}^T$ satisfies :*

$$\sum_{t=1}^T \ell_t(\hat{\mu}_t) - \sum_{t=1}^T \ell_t(\mu_t) \leq \frac{G^2}{2} \sum_{t=1}^T \eta_t - \sum_{t=1}^T \frac{\text{Perf}(t, \hat{\mu}, \mu)}{\eta_t}.$$

*Proof.* The proof is roughly the same than the one for the previous bound. We remark that thanks to strong convexity, one now has :

$$\sum_{t=1}^T \ell_t \left( \hat{\mu}_t \right) - \sum_{t=1}^T \ell_t \left( \mu_t \right) \leqslant \sum_{t=1}^T \langle \nabla \ell_t \left( \hat{\mu}_t \right), \hat{\mu}_t - \mu_t \rangle - \lambda \|\hat{\mu}_t - \mu_t\|^2$$

So the arguments of the previous proof provide us:

$$\sum_{t=1}^T \ell_t(\hat{\mu}_t) - \sum_{t=1}^T \ell_t(\mu_t) \leq \frac{1}{2} \sum_{t=1}^T \left( \frac{1}{\eta_t} - \lambda \right) \|\hat{\mu}_t - \mu_t\|^2 - \frac{\|\hat{\mu}_{t+1} - \hat{\mu}_{t+1}\|^2}{\eta_t}$$

$$+ \sum_{t=1}^T \frac{\eta_t \|\nabla \ell_t \left( \hat{\mu}_t \right)\|^2}{2} - \frac{\text{Perf}(t, \hat{\mu}, \mu)}{\eta_t}$$

$$\leq \frac{1}{2} \sum_{t=1}^T \frac{\|\hat{\mu}_t - \mu_t\|^2}{\eta_{t-1}} - \frac{\|\hat{\mu}_{t+1} - \hat{\mu}_{t+1}\|^2}{\eta_t}$$

$$+ \sum_{t=1}^T \frac{\eta_t \|\nabla \ell_t \left( \hat{\mu}_t \right)\|^2}{2} - \frac{\text{Perf}(t, \hat{\mu}, \mu)}{\eta_t}$$

A telescopic argument and bound over the gradients provides us the final result.

$\square$

**Remark C.3.** *We focus in three specific cases where performance can be linked to classical quantities:*

- *First is just a remark : we totally recover the classical OGD bound for static regret when one has $\mu_{t+1} = \mu_t$ for any $t$.*

- *Second, if our OGD predicts well the minimiser $\mu^*$ after a certain time, i.e. for $t \geq t_0$, $\hat{\mu}_{t+1} \approx \mu^*_{t+1}$. Then one has*

$$\sum_{t=1}^{T} \text{Perf}(t, \hat{\mu}, \mu) \approx -\frac{1}{2} \sum_{t=1}^{T} \frac{||\mu^*_{t+1} - \mu^*_t||^2}{\eta_t} \leq -\frac{1}{\eta_1} S^*_T.$$

*so our result ensures that in this case, OGD has been able to tame the geometry induced by the $\ell_t$s to generate a momentum greater than $S^*_T / \eta_1$*

- *Finally let us consider the overfitting case i.e, for each $t$, $\hat{\mu}_{t+1} \approx \mu^*_t$. Then:*

$$\sum_{t=1}^{T} \text{Perf}(t, \hat{\mu}, \mu) \approx -\frac{1}{2} \sum_{t=1}^{T} \frac{||\mu^*_{t+1} - \mu^*_t||^2}{\eta_t} \leq \frac{1}{\eta_T} S^*_T.$$

*So overfitting will penalise our OGD with at most a factor $S_T / \eta_T$*

However, even if our bounds gives us an intuition on how is the OGD interacting with its environment. One cannot control it directly. If we assume having additional information at each time steps, this notion of performance can help us to enhance OGD.

# D   Proofs of deterministic results

In this section we use the shortcut $\nabla_t := \nabla \ell_t(\hat{\mu}_t)$.

## D.1   A general route of proof

We exhibit in Eq. (1) a general pattern of proof we use several times in this work to bound the dynamic regret. This pattern also structures this document.

$$\sum_{t=1}^{T} \ell_t(\hat{\mu}_t) - \sum_{t=1}^{T} \ell_t(\mu^*_t) = \underbrace{\sum_{t=1}^{T} \ell_t(\hat{\mu}_t) - \sum_{t=1}^{T} \ell_t(\nu_t)}_{=(A)} + \underbrace{\sum_{t=1}^{T} \ell_t(\nu_t) - \sum_{t=1}^{T} \ell_t(\nu_{t+1})}_{=(B)} + \underbrace{\sum_{t=1}^{T} \ell_t(\nu_{t+1}) - \sum_{t=1}^{T} \ell_t(\mu^*_t)}_{=(C)}. \quad (1)$$

Those terms are dealt as follows.

- (A) is controlled by the effect of ADJUST on OGD,ONS,Adagrad. It allows to transform the static guarantees of those algorithms (as stated in Hazan, 2019) into dynamic ones.

- (B) is controlled by the convexity assumptions made on the $\ell_t$s and involve terms like $P_T, S_T$.

- (C) is handled by the way we designed $\nu$.

Our proofs in the rest of this section are based on this general scheme.

### D.2 Proof of Thm 3.1

**Proposition D.1.** *The sequence of predictors $(\hat{\mu}_t)_{t \geq 0}$ obtained through DOGD on a closed convex $\mathcal{K}$ with steps $\eta = (\eta_t)_{t=1..T)}$ with regards to the additional informations $\nu = (\nu_t)_{t=0..T} \in \mathcal{K}^T$ satisfies :*

$$\sum_{t=1}^{T} \ell_t(\hat{\mu}_t) - \sum_{t=1}^{T} \ell_t(\nu_t) \leq \frac{D^2}{2\eta_T} + \frac{G^2}{2} \sum_{t=1}^{T} \eta_t.$$

*Proof.* We fix $t \geq 0$. For the sake of clarity, we rename $\hat{\mu}_{temp} := \hat{\mu}_{temp,t+1} = \hat{\mu}_t - \eta_t \nabla \ell_t(\hat{\mu}_t)$ (where $\hat{\mu}_{temp,t+1}$ is defined in algorithm 6).

Thanks to convexity of the losses, one has:

$$\sum_{t=1}^{T} \ell_t(\hat{\mu}_t) - \sum_{t=1}^{T} \ell_t(\nu_t) \leq \sum_{t=1}^{T} \langle \nabla \ell_t(\hat{\mu}_t), \hat{\mu}_t - \nu_t \rangle.$$

To control this last sum, our intermediary goal is now to control $||\hat{\mu}_{t+1} - \nu_{t+1}||^2$ in function of $\|\hat{\mu}_t - \nu_t\|^2$. To do so, we first exploit Lemma 2.3 which stipulates that $||\hat{\mu}_{t+1} - \nu_{t+1}||^2 \leq \|\hat{\mu}_{temp} - \nu_t\|^2$. Then we control $\langle \nabla \ell_t(\hat{\mu}_t), \hat{\mu}_t - \nu_t \rangle$.

One has:

$$\begin{aligned}
||\hat{\mu}_{t+1} - \nu_{t+1}||^2 &\leq \|\hat{\mu}_{temp} - \nu_t\|^2 \\
&= \|\hat{\mu}_t - \eta_t \nabla \ell_t(\hat{\mu}_t) - \nu_t\|^2 \\
&= \|\hat{\mu}_t - \nu_t\|^2 - 2\eta_t \langle \nabla \ell_t(\hat{\mu}_t), \hat{\mu}_t - \nu_t \rangle + \eta_t^2 \|\nabla \ell_t(\hat{\mu}_t)\|^2
\end{aligned}$$

Hence:

$$\|\hat{\mu}_{t+1} - \nu_{t+1}\|^2 \leqslant \|\hat{\mu}_t - \nu_t\|^2 - 2\eta_t \langle \nabla \ell_t(\hat{\mu}_t), \hat{\mu}_t - \nu_t \rangle + \eta_t^2 G^2.$$

So:

$$\langle \nabla \ell_t(\hat{\mu}_t), \hat{\mu}_t - \nu_t \rangle \leqslant \frac{\|\hat{\mu}_t - \nu_t\|^2 - \|\hat{\mu}_{t+1} - \nu_{t+1}\|^2}{2\eta_t} + \frac{\eta_t G^2}{2}.$$

Summing on $t$, gives

$$\begin{aligned}
\sum_{t=1}^{T} \ell_t(\hat{\mu}_t) - \sum_{t=1}^{T} \ell_t(\nu_t) &\leq \sum_{t=1}^{T} \|\hat{\mu}_t - \nu_t\|^2 \left( \frac{1}{2\eta_t} - \frac{1}{2\eta_{t-1}} \right) + \frac{\eta_t G^2}{2} \\
&\leq \frac{D^2}{2\eta_T} + \frac{G^2}{2} \sum_{t=1}^{T} \eta_t.
\end{aligned}$$

Hence the final result.

$\square$

Now we are able to prove our result:

**Proof of Thm. 3.1**

*Proof.* We control the terms presented in Eq. (1). proposition D.1 ensures us that:

$$(A) \leq \frac{D^2}{2\eta_T} + \frac{G^2}{2} \sum_{t=1}^{T} \eta_t$$

$$\leq \frac{3}{2} GD\sqrt{T},$$

The last line holding thanks to the definition of $\eta$ and that $\sum_{t=1}^{T} \frac{1}{\sqrt{t}} \leq 2\sqrt{T}$.

We now have to deal with (B) and (C) of Eq. (1).

(B) is handled using the strong convexity of $\ell_t$ for any $t$ :

$$\ell_t(\nu_t) - \ell_t(\nu_{t+1}) \leq \nabla \ell_t(\nu_t)^\top (\nu_t - \nu_{t+1}) - \lambda ||\nu_{t+1} - \nu_t||^2$$
$$\leq ||\nabla \ell_t(\nu_t)||.||\nu_{t+1} - \nu_t|| - \lambda ||\nu_{t+1} - \nu_t||^2 \qquad \text{Cauchy-Schwarz}$$
$$\leq G||\nu_{t+1} - \nu_t|| - \lambda ||\nu_{t+1} - \nu_t||^2.$$

Summing over all $t$ gives us :

$$(B) \leq GP_T(\nu) - \lambda S_T(\nu).$$

To deal with (C), we exploit Lemma 2.5. Indeed, our choice of steps ensure us that at each step $j$: $\frac{1}{\eta'_j} - \lambda = \lambda(j-1) = \frac{1}{\eta'_{j-1}}$. We have at each time $t$:

$$\ell_t(\nu_{t+1}) - \ell_t(\mu_t^*) \leq \frac{G^2}{K} \sum_{j=1}^{K} \eta'_j = \frac{G^2}{\lambda K} \sum_{j=1}^{K} \frac{1}{j}$$
$$\leq \frac{G^2(1 + \log(K))}{\lambda K}.$$

Finally:

$$(C) \leq T \frac{G^2(1 + \log(K)}{\lambda K}$$
$$\leq \frac{G^2}{\lambda} \sqrt{T}(1 + \log(1 + T))$$

The last line holding because $K = \lceil \sqrt{T} \rceil$.

Combining the bounds of (A),(B),(C) concludes the proof.

$\square$

## D.3   Proof of Thm 3.3

We need first to introduce on exp-concave funtion.

**Definition D.2.** *A function $f : \mathbb{R}^n \to \mathbb{R}$ is $\alpha$ exp-concave over a convex $\mathcal{K}$ if the function $g = \exp(-\alpha f)$ is concave on $\mathcal{K}$.*

One also recalls the following lemma coming from (Hazan, 2019, Lemma 4.3)

**Lemma D.3.** *Let $f : \mathcal{K} \to \mathbb{R}$ be an $\alpha$-exp-concave function, and $D, G$ denote the diameter of $\mathcal{K}$ and a bound on the (sub)gradients of $f$ respectively. The following holds for all $\gamma \leq \frac{1}{2} \min \left\{ \frac{1}{4GD}, \alpha \right\}$ and all $\mathbf{x}, \mathbf{y} \in \mathcal{K}$ :*

$$f(\mathbf{x}) \geq f(\mathbf{y}) + \nabla f(\mathbf{y})^\top (\mathbf{x} - \mathbf{y}) + \frac{\gamma}{2} (\mathbf{x} - \mathbf{y})^\top \nabla f(\mathbf{y}) \nabla f(\mathbf{y})^\top (\mathbf{x} - \mathbf{y}).$$

One now states a key preliminary result of this section (proposition D.4) whoch exploits the exp-concavity property.

**Proposition D.4.** *We assume our loss functions $\ell_t$ to be $\alpha$ exp-concave. Let $\{\hat{\mu}_t\}$ being the output of D-ONS (algorithm 4) with $\gamma = \frac{1}{2} \min \left\{ \frac{1}{GD}, \alpha \right\}$, $\varepsilon = \frac{1}{\gamma^2 D^2}$. We then have, for $T > 4$ and any additional knowledge $\nu$:*

$$\sum_{t=1}^{T} \ell_t(\hat{\mu}_t) - \ell_t(\nu_t) \leq 2 \left( \frac{1}{\alpha} + GD \right) d \log(T).$$

*Proof.* The proof is similar to the one of (Hazan, 2019, Thm 4.5) which holds for static regret. We prove Lemma D.5 which is an adaptation of (Hazan, 2019, Lemma 4.6).

**Lemma D.5.** *Let $\{\hat{\mu}_t\}$ being the output of algorithm 4 with $\gamma = \frac{1}{2} \min \left\{ \frac{1}{GD}, \alpha \right\}$, $\varepsilon = \frac{1}{\gamma^2 D^2}$. We then have, for $T > 4$ and any additional knowledge $\nu$:*

$$\sum_{t=1}^{T} \ell_t(\hat{\mu}_t) - \ell_t(\nu_t) \leq \left( \frac{1}{\alpha} + GD \right) \left( 1 + \sum_{t=1}^{T} \nabla_t A_t^{-1} \nabla_t^\top \right).$$

*Proof.* We fix $t \geq 1$ and we first apply Lemma D.3:

$$\ell_t(\hat{\mu}_t) - \ell_t(\nu_t) \leq \nabla_t^\top (\hat{\mu}_t - \nu_t) - \frac{\gamma}{2} (\hat{\mu}_t - \nu_t)^\top \nabla_t \nabla_t^\top (\hat{\mu}_t - \nu_t)$$

Recalling the definition of $\hat{\mu}_{temp,t+1}$, substracting by $\nu_t$ and multiplying by $A_t$ gives us:

$$\hat{\mu}_{temp,t+1} - \nu_t = \hat{\mu}_t - \nu_t - \frac{1}{\gamma} A_t^{-1} \nabla_t \qquad (2)$$

and:

$$A_t \left( \hat{\mu}_{temp,t+1} - \nu_t \right) = A_t \left( \hat{\mu}_t - \nu_t \right) - \frac{1}{\gamma} \nabla_t \qquad (3)$$

Multiplying the transpose of Eq. (2) by Eq. (3) gives us:

$$\left( \hat{\mu}_{temp,t+1} - \nu_t \right)^\top A_t \left( \hat{\mu}_{temp,t+1} - \nu_t \right) = \left( \hat{\mu}_t - \nu_t \right)^\top A_t \left( \hat{\mu}_t - \nu_t \right) - \frac{2}{\gamma} \nabla_t^\top \left( \hat{\mu}_t - \nu_t \right) + \frac{1}{\gamma^2} \nabla_t^\top A_t^{-1} \nabla_t. \qquad (4)$$

Our goal is to lower bound the term on left hand-side of this equality. To do so, we first remark

$$\left( \hat{\mu}_{temp,t+1} - \nu_t \right)^\top A_t \left( \hat{\mu}_{temp,t+1} - \nu_t \right) \quad = \| \hat{\mu}_{temp,t+1} - \nu_t \|_{A_t}^2$$

Because $A_t$ is a positive definite matrix, Lemma 2.3 holds, which allows us to say that $\| \hat{\mu}_{temp,t+1} - \nu_t \|_{A_t}^2 \geq \| \hat{\mu}_{t+1} - \nu_{t+1} \|_{A_t}^2$. Thus:

$$\left(\hat{\mu}_{temp,t+1} - \nu_t\right)^\top A_t \left(\hat{\mu}_{temp,t+1} - \nu_t\right) \geq \left\|\hat{\mu}_{t+1} - \nu_{t+1}\right\|_{A_t}^2$$
$$= \left(\hat{\mu}_{t+1} - \nu_{t+1}\right)^\top A_t \left(\hat{\mu}_{t+1} - \nu_{t+1}\right)$$

This fact together with Eq. (4) gives:

$$\nabla_t^\top \left(\hat{\mu}_t - \nu_t\right) \leq \frac{1}{2\gamma} \nabla_t^\top A_t^{-1} \nabla_t + \frac{\gamma}{2} \left(\hat{\mu}_t - \nu_t\right)^\top A_t \left(\hat{\mu}_t - \nu_t\right)$$
$$- \frac{\gamma}{2} \left(\hat{\mu}_{t+1} - \nu_{t+1}\right)^\top A_t \left(\hat{\mu}_{t+1} - \nu_{t+1}\right).$$

Now, summing up over $t = 1$ to $T$ we get that

$$\sum_{t=1}^T \nabla_t^\top \left(\hat{\mu}_t - \nu_t\right) \leq \frac{1}{2\gamma} \sum_{t=1}^T \nabla_t^\top A_t^{-1} \nabla_t + \frac{\gamma}{2} \left(\mu_1 - \nu_1\right)^\top A_1 \left(\mu_1 - \nu_1\right)$$
$$+ \frac{\gamma}{2} \sum_{t=2}^T \left(\hat{\mu}_t - \nu_t\right)^\top \left(A_t - A_{t-1}\right) \left(\hat{\mu}_t - \nu_t\right)$$
$$- \frac{\gamma}{2} \left(\hat{\mu}_{T+1} - \nu_{T+1}\right)^\top A_T \left(\hat{\mu}_{T+1} - \nu_{T+1}\right)$$
$$\leq \frac{1}{2\gamma} \sum_{t=1}^T \nabla_t^\top A_t^{-1} \nabla_t + \frac{\gamma}{2} \sum_{t=1}^T \left(\hat{\mu}_t - \nu_t\right)^\top \nabla_t \nabla_t^\top \left(\hat{\mu}_t - \nu_t\right)$$
$$+ \frac{\gamma}{2} \left(\mu_1 - \nu_1\right)^\top \left(A_1 - \nabla_1 \nabla_1^\top\right) \left(\mu_1 - \nu_1\right)$$

In the last inequality we use the fact that $A_t - A_{t-1} = \nabla_t \nabla_t^\top$, and the fact that the matrix $A_T$ is PSD to bound the last term before the inequality by 0. Thus,

$$\sum_{t=1}^T \ell_t(\hat{\mu}_t) - \ell_t(\nu_t) \leq \frac{1}{2\gamma} \sum_{t=1}^T \nabla_t^\top A_t^{-1} \nabla_t + \frac{\gamma}{2} \left(\mu_1 - \nu_1\right)^\top \left(A_1 - \nabla_1 \nabla_1^\top\right) \left(\mu_1 - \nu_1\right)$$

Using that $A_1 - \nabla_1 \nabla_1^\top = \varepsilon I_n$, $\varepsilon = \frac{1}{\gamma^2 D^2}$ and that $\mathcal{K}$ has a finite diameter $D$ gives us :

$$\sum_{t=1}^T \ell_t(\hat{\mu}_t) - \ell_t(\nu_t) \leq \frac{1}{2\gamma} \sum_{t=1}^T \nabla_t^\top A_t^{-1} \nabla_t + \frac{\gamma}{2} D^2 \varepsilon$$
$$\leq \frac{1}{2\gamma} \sum_{t=1}^T \nabla_t^\top A_t^{-1} \nabla_t + \frac{1}{2\gamma}$$

Since $\gamma = \frac{1}{2} \min \left\{\frac{1}{GD}, \alpha\right\}$, we have $\frac{1}{\gamma} \leq 2 \left(\frac{1}{\alpha} + GD\right)$. This gives the lemma.

$$\square$$

The rest of the proof now follows the exact same route than (Hazan, 2019, Thm 4.5).

**Proof of proposition D.4** First we show that the term $\sum_{t=1}^T \nabla_t^\top A_t^{-1} \nabla_t$ is upper bounded by a telescoping sum. Notice that
$$\nabla_t^\top A_t^{-1} \nabla_t = A_t^{-1} \bullet \nabla_t \nabla_t^\top = A_t^{-1} \bullet \left(A_t - A_{t-1}\right)$$
where for matrices $A, B \in \mathbb{R}^{n \times n}$ we denote by $A \bullet B = \sum_{i=1}^n \sum_{j=1}^n A_{ij} B_{ij} = \text{Tr}\left(AB^\top\right)$, which is equivalent to the inner product of these matrices as vectors in $\mathbb{R}^{n^2}$.

For real numbers $a, b \in \mathbb{R}_+$, the first order Taylor expansion of the logarithm of $b$ at $a$ implies $a^{-1}(a - b) \leq \log \frac{a}{b}$. An analogous fact holds for positive semidefinite matrices, i.e., $A^{-1} \bullet (A - B) \leq \log \frac{|A|}{|B|}$, where $|A|$

denotes the determinant of the matrix $A$ (this is proved in Hazan, 2019, Lemma 4.7). Using this fact we have

$$\sum_{t=1}^{T} \nabla_t^\top A_t^{-1} \nabla_t = \sum_{t=1}^{T} A_t^{-1} \bullet \nabla_t \nabla_t^\top$$

$$= \sum_{t=1}^{T} A_t^{-1} \bullet (A_t - A_{t-1})$$

$$\leq \sum_{t=1}^{T} \log \frac{|A_t|}{|A_{t-1}|} = \log \frac{|A_T|}{|A_0|}$$

Since $A_T = \sum_{t=1}^{T} \nabla_t \nabla_t^\top + \varepsilon I_n$ and $\|\nabla_t\| \leq G$, the largest eigenvalue of $A_T$ is at most $TG^2 + \varepsilon$. Hence the determinant of $A_T$ can be bounded by $|A_T| \leq (TG^2 + \varepsilon)^d$. Hence recalling that $\varepsilon = \frac{1}{\gamma^2 D^2}$ and $\gamma = \frac{1}{2} \min\left\{\frac{1}{GD}, \alpha\right\}$, for $T > 4$

$$\sum_{t=1}^{T} \nabla_t^\top A_t^{-1} \nabla_t \leq \log \left(\frac{TG^2 + \varepsilon}{\varepsilon}\right)^d \leq d \log \left(TG^2 \gamma^2 D^2 + 1\right) \leq d \log T$$

Plugging into Lemma D.5 we obtain

$$\sum_{t=1}^{T} \ell_t(\hat{\mu}_t) - \ell_t(\nu_t) \leq \left(\frac{1}{\alpha} + GD\right)(d \log T + 1)$$

which implies the theorem for $d > 1, T \geq 4$.

$\square$

We now can prove Thm. 3.3.

**Proof of Thm. 3.3.**

*Proof.* We control the terms presented in Eq. (1). To deal with (A), we exploit proposition D.4 knowing that a $\lambda$-strongly convex function with its gradient bounded by $G$ is $\lambda/G^2$ exp-concave:

$$(A) \leq 2 \left(\frac{G^2}{\lambda} + GD\right) d(1 + \log(T))$$

We now have to deal with (B) and (C) of Eq. (1).

(B) is handled using the strong convexity of $\ell_t$ for any $t$ :

$$\begin{aligned} \ell_t(\nu_t) - \ell_t(\nu_{t+1}) &\leq \nabla \ell_t(\nu_t)^\top (\nu_t - \nu_{t+1}) - \lambda \|\nu_{t+1} - \nu_t\|^2 \\ &\leq \|\nabla \ell_t(\nu_t)\| . \|\nu_{t+1} - \nu_t\| - \lambda \|\nu_{t+1} - \nu_t\|^2 \qquad \text{Cauchy-Schwarz} \\ &\leq G \|\nu_{t+1} - \nu_t\| - \lambda \|\nu_{t+1} - \nu_t\|^2. \end{aligned}$$

Summing over all $t$ gives us :

$$(B) \leq G P_T(\nu) - \lambda S_T(\nu).$$

To deal with (C), we exploit Lemma 2.5. Indeed, our choice of steps ensure us that at each step $j$: $\frac{1}{\eta'_j} - \lambda = \lambda(j-1) = \frac{1}{\eta'_{j-1}}$. We have at each time $t$:

$$\ell_t(\nu_{t+1}) - \ell_t(\mu^*_t) \leq \frac{G^2}{K} \sum_{j=1}^{K} \eta'_j = \frac{G^2}{\lambda K} \sum_{j=1}^{K} \frac{1}{j}$$
$$\leq \frac{G^2(1 + \log(K))}{\lambda K}.$$

Finally:

$$(C) \leq T \frac{G^2(1 + \log(K))}{\lambda K}$$
$$= \frac{G^2}{\lambda}(1 + \log(T))$$

The last line holding because $K = T$.

Combining the bounds on (A),(B),(C) concludes the proof.

$\square$

### D.4 Proof of Thm 3.5

We first start with a key result for our study of dynamic Adagrad.

**Proposition D.6.** *We assume our loss functions $\ell_t$ to be convex. Let $\{\hat{\mu}_t\}$ being the output of D-Adagrad (algorithm 5) with $\varepsilon = \frac{2}{D^2}, \eta = \frac{D}{\sqrt{2}}$. We then have, for any additional knowledge $\nu$:*

$$\sum_{t=1}^{T} \ell_t(\hat{\mu}_t) - \ell_t(\nu_t) \leq \sqrt{2}D \left(1 + \sqrt{\min_{H \in \mathcal{H}} \sum_t \|\nabla_t\|_H^{*2}}\right)$$

*where $\mathcal{H} := \{X \in \mathbb{R}^{n \times n} \mid Tr(X) \leq 1, X \succeq 0\}$ and for a fixed $H$, $\|\mu\|_H^{*2} = \mu^T H^{-1} \mu$ where $H^{-1}$ refers to the Moore-Penrose pseudoinverse.*

*Proof.* The proof follows the route of (Hazan, 2019, Thm 5.12) for the full-matrix version of Adagrad. As for dynamic ONS, our only work consists in modifying a lemma of Hazan's proof (Hazan, 2019, Lemma 5.13), the rest holding similarly.

For the sake of completeness, we state all the lemma of interest in this proof, most of them are directly extracted from (Hazan, 2019, Sec.5.6). We start with (Hazan, 2019, Lemma 11).

**Lemma D.7.** *For $H_T$ the last output of Adagrad, we have*

$$\sqrt{\min_{H \in \mathcal{H}} \sum_t \|\nabla_t\|_H^{*2}} = \boldsymbol{Tr}(H_T)$$

We present now our lemma of interest (Hazan, 2019, Lemma 5.13)

**Lemma D.8.**

$$\sum_{t=1}^{T} \ell_t(\hat{\mu}_t) - \ell_t(\nu_t) \leq 2D + \frac{\eta}{2}\left(G_T \bullet H_T^{-1} + \text{Tr}(H_T)\right) + \frac{1}{2\eta} \sum_{t=1}^{T} (\hat{\mu}_t - \hat{\nu}_t)^\top (H_t - H_{t-1})(\hat{\mu}_t - \nu_t).$$

*Proof.* First, recall that $\sum_{t=1}^{T} \ell_t(\hat{\mu}_t) - \ell_t(\nu_t) \leq \sum_{t=1}^{T} \nabla_t^{\top} (\hat{\mu}_t - \nu_t)$.

By the definition of $\hat{\mu}_{temp,t+1}$ :

$$\hat{\mu}_{temp,t+1} - \nu_t = \hat{\mu}_t - \nu_t - \eta H_t^{-1} \nabla_t \tag{5}$$

and multipying by $H_t$ gives:

$$H_t \left( \hat{\mu}_{temp,t+1} - \nu_t \right) = H_t \left( \hat{\mu}_t - \nu_t \right) - \eta \nabla_t. \tag{6}$$

Multiplying the transpose of Eq. (5) by Eq. (6) we get

$$\left( \hat{\mu}_{temp,t+1} - \nu_t \right)^{\top} H_t \left( \hat{\mu}_{temp,t+1} - \nu_t \right)$$
$$= \left( \hat{\mu}_t - \nu_t \right)^{\top} H_t \left( \hat{\mu}_t - \nu_t \right) - 2\eta \nabla_t^{\top} \left( \hat{\mu}_t - \nu_t \right) + \eta^2 \nabla_t^{\top} H_t^{-1} \nabla_t. \tag{7}$$

Focusing on the left-hand side of the equality, one remarks that:

$$\left( \hat{\mu}_{temp,t+1} - \nu_t \right)^{\top} H_t \left( \hat{\mu}_{temp,t+1} - \nu_t \right) = \| \hat{\mu}_{temp,t+1} - \nu_t \|_{H_t}^2$$

Since $H_t$ is a PD matrix, one can apply Lemma 2.3 to obtain that $\| \hat{\mu}_{t+1} - \nu_{t+1} \|_{H_t}^2 \leq \| \hat{\mu}_{temp,t+1} - \nu_t \|_{H_t}^2$. Applying this result gives:

$$\left( \hat{\mu}_{temp,t+1} - \nu_t \right)^{\top} H_t \left( \hat{\mu}_{temp,t+1} - \nu_t \right) \geq \| \hat{\mu}_{t+1} - \nu_{t+1} \|_{H_t}^2$$

This fact together with Eq. (7) gives

$$\nabla_t^{\top} \left( \hat{\mu}_t - \nu_t \right) \leq \frac{\eta}{2} \nabla_t^{\top} H_t^{-1} \nabla_t + \frac{1}{2\eta} \left( \| \hat{\mu}_t - \nu_t \|_{H_t}^2 - \| \hat{\mu}_{t+1} - \nu_{t+1} \|_{H_t}^2 \right)$$

Now, summing up over $t = 1$ to $T$ we get that

$$\sum_{t=1}^{T} \nabla_t^{\top} \left( \hat{\mu}_t - \nu_t \right) \leq$$

$$\frac{\eta}{2} \sum_{t=1}^{T} \nabla_t^{\top} H_t^{-1} \nabla_t + \frac{1}{2\eta} \| \mu_1 - \nu_1 \|_{H_0}^2 + \frac{1}{2\eta} \sum_{t=1}^{T} \left( \| \hat{\mu}_t - \nu_t \|_{H_t}^2 - \| \hat{\mu}_t - \nu_t \|_{H_{t-1}}^2 \right) - \frac{1}{2\eta} \| \hat{\mu}_{t+1} - \nu_{t+1} \|_{H_T}^2$$

$$\leq \frac{\eta}{2} \sum_{t=1}^{T} \nabla_t^{\top} H_t^{-1} \nabla_t + \sqrt{2} D + \frac{1}{2\eta} \sum_{t=1}^{T} \left( \hat{\mu}_t - \nu_t \right)^{\top} \left( H_t - H_{t-1} \right) \left( \hat{\mu}_t - \nu_t \right).$$

In the last inequality we used the fact that $\varepsilon = \frac{2}{D^2}$ and bounded $\| \mu_1 - \nu_1 \|$ by $D^2$ .

We now prove that $\sum_{t=1}^{T} \nabla_t^{\top} H_t^{-1} \nabla_t \leq \left( G_T \bullet H_T^{-1} + \text{Tr}(H_T) \right)$. To this end, define the functions

$$\Psi_t(H) = \nabla_t \nabla_t^{\top} \bullet H^{-1}, \Psi_0(H) = \text{Tr}(H).$$

By definition, $H_t$ is the minimizer of $\sum_{i=0}^{t} \Psi_i$ over $\mathcal{H}$ which can be related to a FTL strategy. Thus, using (Hazan, 2019, Lemma 5.4), we have that

$$\sum_{t=1}^{T} \nabla_t^{\top} H_t^{-1} \nabla_t = \sum_{t=1}^{T} \Psi_t \left( H_t \right)$$

$$\leq \sum_{t=1}^{T} \Psi_t \left( H_T \right) + \Psi_0 \left( H_T \right) - \Psi_0 \left( H_0 \right)$$

$$= G_T \bullet H_T^{-1} + \text{Tr} \left( H_T \right)$$

This concludes the proof □

Lemma D.8 gives us two terms to be bounded. To do so, we use (Hazan, 2019, Lemmas 5.14,5.15) to conclude the proof. Those lemmas are gathered below.

**Lemma D.9.** *For algorithm 5, the following holds*

$$G_T \bullet H_T^{-1} \leq \text{Tr}\left(H_T\right).$$

**Lemma D.10.** *Recall that $D$ the Euclidean diameter of $\mathcal{K}$. Then the following bound holds,* $\sum_{t=1}^{T} \|\mathbf{x}_t - \mathbf{x}^\star\|_{H_t - H_{t-1}}^2 \leq D^2 \text{Tr}\left(H_T\right).$

Now combining Lemma D.8 with the above two lemmas, and using $\eta = \frac{D}{\sqrt{2}}$ appropriately, we obtain the theorem.

$\square$

We now can prove Thm. 3.5.

**Proof of Thm. 3.5.**

*Proof.* We control the terms presented in Eq. (1). To deal with (A), we exploit proposition D.6:

$$(A) \leq \sqrt{2}D \left(1 + \sqrt{\min_{H \in \mathcal{H}} \sum_t \|\nabla_t\|_H^{*2}}\right)$$

We now have to deal with (B) and (C) of Eq. (1).

(B) is handled using the strong convexity of $\ell_t$ for any $t$ :

$$
\begin{aligned}
\ell_t(\nu_t) - \ell_t(\nu_{t+1}) &\leq \nabla \ell_t(\nu_t)^\top (\nu_t - \nu_{t+1}) - \lambda \|\nu_{t+1} - \nu_t\|^2 \\
&\leq \|\nabla \ell_t(\nu_t)\|.\|\nu_{t+1} - \nu_t\| - \lambda \|\nu_{t+1} - \nu_t\|^2 \qquad \text{Cauchy-Schwarz} \\
&\leq G\|\nu_{t+1} - \nu_t\| - \lambda \|\nu_{t+1} - \nu_t\|^2.
\end{aligned}
$$

Summing over all $t$ gives us :

$$(B) \leq G P_T(\nu) - \lambda S_T(\nu).$$

To deal with (C), we exploit Lemma 2.5. Indeed, our choice of steps ensure us that at each step $j$: $\frac{1}{\eta_j'} - \lambda = \lambda(j-1) = \frac{1}{\eta_{j-1}'}$. We have at each time $t$:

$$
\begin{aligned}
\ell_t(\nu_{t+1}) - \ell_t(\mu_t^*) &\leq \frac{G^2}{K} \sum_{j=1}^{K} \eta_j' = \frac{G^2}{\lambda K} \sum_{j=1}^{K} \frac{1}{j} \\
&\leq \frac{G^2(1 + \log(K))}{\lambda K}.
\end{aligned}
$$

Finally:

$$
\begin{aligned}
(C) &\leq T \frac{G^2(1 + \log(K))}{\lambda K} \\
&= \frac{G^2}{\lambda}(1 + \log(T))
\end{aligned}
$$

The last line holding because $K = T$.

Combining the bounds on (A),(B),(C) concludes the proof.

$\square$

# E   Proofs of probabilistic results

## E.1   The SOCO framework

In what follows, for a certain filtration $(\mathcal{F}_t)_{t \geq 1}$, we denote by $\mathbb{E}_{t-1}[.] := \mathbb{E}[. \mid \mathcal{F}_{t-1}]$. SOCO's framework has been introduced in Wintenberger (2021). It focuses on a more general notion of regret presented below.

**Definition E.1.** *For loss function $\ell_t$, we denote by $(\mathcal{F}_t)_t$ a filtration s.t. $\ell_t$ is $\mathcal{F}_t$-measurable. For some predictors $(\hat{\mu}_t)_{t=1..T} \in \mathcal{K}$ we define the* dynamic averaged regret *with regards to $(\mu_t)_{t=1..T} \in \mathcal{K}^T$ as follows:*

$$D\text{-}Av\text{-}Regret_T := \sum_{t=1}^{T} \mathbb{E}_{t-1}[\ell_t(\hat{\mu}_t)] - \sum_{t=1}^{T} \mathbb{E}_{t-1}[\ell_t(\mu_t)].$$

We use SOCO here with the two following assumptions:

**(H1)**   The diameter of $\mathcal{K}$ is $D < \infty$ so that $\|x - y\| \leq D, x, y \in \mathcal{K}$, and the functions $\ell_t$ are continuously differentiable over $\mathcal{K}$ a.s. and the gradients are bounded by $G < \infty : \sup_{x \in \mathcal{K}} \|\nabla \ell_t(x)\| \leq G$ a.s.,$t \geq 1$

**(H2)**   The random loss functions $(\ell_t)$ are stochastically exp-concave i.e. it exists $\alpha > 0$ such that, for any $\mu_1, \mu_2 \in \mathcal{K}$:

$$\mathbb{E}_{t-1}[\ell_t(\mu_2)] \leq \mathbb{E}_{t-1}[\ell_t(\mu_1)] + \mathbb{E}_{t-1}[\nabla \ell_t(\mu_2)^T(\mu_2 - \mu_1)] - \frac{\alpha}{2}\mathbb{E}_{t-1}\left[\left(\nabla \ell_t(\mu_2)^T(\mu_2 - \mu_1)\right)^2\right], \quad x, y \in \mathcal{K}.$$

**Remark E.2.** *A $\lambda$-strongly convex function with its gradients bounded by $G$ in absolute value is $\alpha$ stochastically exp-concave with $\alpha = \lambda/G^2$*

Note that Prop 3 of SOCO is valid for dynamic regret:

**Lemma E.3** ((Wintenberger 2021, Proposition 3)). *For any decision sequence $(\hat{\mu}_t)_t \in \mathcal{K}^T, (\mu_t)_t \in (\mathcal{K}^T)^2$, under (H1) and (H2), with probability $1 - \delta$, it holds for any $\beta > 0$ and any $T \geq 1$*

$$\sum_{t=1}^{T} \mathbb{E}_{t-1}[\ell_t(\hat{\mu}_t)] - \sum_{t=1}^{T} \mathbb{E}_{t-1}[\ell_t(\mu_t)] \leq \sum_{t=1}^{T} \nabla \ell_t(\hat{\mu}_t)^T(\hat{\mu}_t - \mu_t)$$
$$+ \frac{\beta}{2}\sum_{t=1}^{T}\left(\nabla \ell_t(\hat{\mu}_t)^T(\hat{\mu}_t - \mu_t)\right)^2 + \frac{2}{\beta}\log\left(\delta^{-1}\right)$$
$$+ \frac{\beta - \alpha}{2}\sum_{t=1}^{T}\mathbb{E}_{t-1}\left[\left(\nabla \ell_t(\hat{\mu}_t)^T(\hat{\mu}_t - \mu_t)\right)^2\right]$$

## E.2   Proof of Thm. 3.2

Our goal is now to combine this property with our dynamic OGD. To do so, we want to control the quadratic terms in Lemma E.3. This is the goal of proposition E.4.

**Proposition E.4.** *For any decision sequence $(\hat{\mu}_t)_t$, any sequence $(\mu_t)_t$ such that for any $t; (\hat{\mu}_t, \mu_t)$ is $\mathcal{F}_{t-1}$-measurable, with probability $1 - 2\delta$, it holds for any $T \geq 1$*

$$\sum_{t=1}^{T} \mathbb{E}_{t-1}[\ell_t(\hat{\mu}_t)] - \sum_{t=1}^{T} \mathbb{E}_{t-1}[\ell_t(\mu_t)] \leq \sum_{t=1}^{T} \nabla \ell_t(\hat{\mu}_t)^T(\hat{\mu}_t - \mu_t) + \left(2(GD)^2 + 6\frac{G^2}{\lambda}\right)\log\left(\delta^{-1}\right)$$

*Proof.* We define $\alpha = \lambda/G^2$ and $Y_t = \nabla \ell_t (\hat{\mu}_t)^T (\hat{\mu}_t - \mu_t)$ remark that $|Y_t| \leq GD$ a.s, we then exploit a corollary of a Poissonian inequality stated in (Wintenberger, 2021, Eq. (7)). With probability $1 - \delta$ we have:

$$\sum_{t=1}^{T} Y_t^2 \leq 2 \sum_{t=1}^{T} \mathbb{E}_{t-1}[Y_t^2] + 2(GD)^2 \log(1/\delta)$$

Thus, taking an union bound to make hold this inequality simultaneously with the one of Lemma E.3 and taking $\beta$ such that $3\beta - \alpha = 0$ gives us with probability $1 - 2\delta$:

$$\sum_{t=1}^{T} \mathbb{E}_{t-1}[\ell_t(\hat{\mu}_t)] - \sum_{t=1}^{T} \mathbb{E}_{t-1}[\ell_t(\mu_t)] \leq \sum_{t=1}^{T} \nabla \ell_t (\hat{\mu}_t)^T (\hat{\mu}_t - \mu_t) + \left( 2(GD)^2 + 6 \frac{G^2}{\lambda} \right) \log \left( \delta^{-1} \right)$$

This concludes the proof. □

We are now able to prove Thm. 3.2:

**Proof of Thm. 3.2.**

*Proof.* We first state that for any $(\hat{\mu}_t, \mu_t)$:

$$
\begin{aligned}
\sum_{t=1}^{T} \mathbb{E}_{t-1}[\ell_t(\hat{\mu}_t)] - \sum_{t=1}^{T} \mathbb{E}_{t-1}[\ell_t(\mu_t)] &= \sum_{t=1}^{T} \mathbb{E}_{t-1} \left[ \ell_t(\hat{\mu}_t) - \ell_t(\mu_t) \right] \\
&\leq \sum_{t=1}^{T} \mathbb{E}_{t-1} \left[ \ell_t(\hat{\mu}_t) - \ell_t(\mu_t^*) \right] \qquad \text{with } \mu_t^* = \mathrm{argmin}_{\mu \in \mathcal{K}} \, \ell_t(\mu) \\
&= \underbrace{\sum_{t=1}^{T} \mathbb{E}_{t-1} \left[ \ell_t(\hat{\mu}_t) - \ell_t(\nu_t) \right]}_{:=S_1} + \underbrace{\sum_{t=1}^{T} \mathbb{E}_{t-1} \left[ \ell_t(\nu_t) - \ell_t(\nu_{t+1}) \right]}_{:=S_2} \\
&\quad + \underbrace{\sum_{t=1}^{T} \mathbb{E}_{t-1} \left[ \ell_t(\nu_{t+1}) - \ell_t(\mu_t^*) \right]}_{:=S_3}
\end{aligned}
$$

The sum $S_1$ is controlled by applying proposition E.4. Then the sum $\sum_{t=1}^{T} \nabla \ell_t(\hat{\mu}_t)^T (\hat{\mu}_t - \nu_t)$ is handled by proposition D.1. We then obtain with our specific choice of steps:

$$S_1 \leq \frac{3}{2} GD\sqrt{T} + \left( 2(GD)^2 + 6 \frac{G^2}{\lambda} \right) \log \left( \delta^{-1} \right) = O(\sqrt{T}).$$

To control the two last sums, we exploit some arguments provided in Thm. 3.1. More precisely we use the bounds designed to control the sum (B) and (C) in the Thm. 3.1's' proof. We then have for any $t \geq 0$, by strong convexity of the losses:

$$\ell_t(\nu_t) - \ell_t(\nu_{t+1}) \leq G||\nu_{t+1} - \nu_t|| - \lambda ||\nu_{t+1} - \nu_t||^2.$$

Also, our choice of steps gives for any $j$: $\frac{1}{\eta'_j} - \lambda = \lambda(j-1) = \frac{1}{\eta'_{j-1}}$. Then, using Lemma 2.5 gives:

$$\ell_t(\nu_{t+1}) - \ell_t(\mu_t^*) \leq \frac{G^2}{K}\sum_{j=1}^{K}\eta'_j = \frac{G^2}{\lambda K}\sum_{j=1}^{K}\frac{1}{j}$$
$$\leq \frac{G^2(1+\log(K))}{\lambda K}.$$

Then, applying our conditional expectations, recalling that $K = \lceil\sqrt{T}\rceil$ and summing over $t$ gives us.

$$S_2 \leq \sum_{t=1}^{T}\mathbb{E}_{t-1}\left[G\|\nu_{t+1} - \nu_t\| - \lambda\|\nu_{t+1} - \nu_t\|^2\right],$$

$$S_3 \leq \frac{G^2}{\lambda}\sqrt{T}(1 + \log(1+T)) = \tilde{O}(\sqrt{T}).$$

To conclude the proof, one remarks that if one defines

$$M_T := \sum_{t=1}^{T}\mathbb{E}_{t-1}\left[G\|\nu_{t+1} - \nu_t\| - \lambda\|\nu_{t+1} - \nu_t\|^2\right] - (GP_T(\nu) - \lambda S_T(\nu))$$

Then:

$$S_2 \leq \sum_{t=1}^{T}\mathbb{E}_{t-1}\left[G\|\nu_{t+1} - \nu_t\| - \lambda\|\nu_{t+1} - \nu_t\|^2\right] = M_T + GP_T(\nu) - \lambda S_T(\nu)$$

$(M_t)_{t\geq 0}$ is a martingale and furthermore for any $t \geq 0$, $-\lambda D^2 \leq \underbrace{G\|\nu_{t+1} - \nu_t\| - \lambda\|\nu_{t+1} - \nu_t\|^2}_{=M_t - M_{t-1}} \leq GD$.

Thus, applying Azuma-Hoeffding's inequality gives us, with probability $1 - \delta$ that $M_T \leq O(\sqrt{T})$

So with probability $1 - \delta$, one has $S_2 \leq GP_T(\nu) - \lambda S_T(\nu) + O(\sqrt{T})$.

Applying an union bound on the bounds of $S_1, S_2$ and summing the bound of $S_1, S_2, S_3$ concludes the proof.

$\square$

### E.3 Proof of Thm. 3.4

*Proof.* We first state that for any $(\hat{\mu}_t, \mu_t)$:

$$\sum_{t=1}^{T}\mathbb{E}_{t-1}[\ell_t(\hat{\mu}_t)] - \sum_{t=1}^{T}\mathbb{E}_{t-1}[\ell_t(\mu_t)] = \sum_{t=1}^{T}\mathbb{E}_{t-1}\left[\ell_t(\hat{\mu}_t) - \ell_t(\mu_t)\right]$$
$$\leq \sum_{t=1}^{T}\mathbb{E}_{t-1}\left[\ell_t(\hat{\mu}_t) - \ell_t(\mu_t^*)\right] \text{ with } \mu_t^* = \text{argmin}_{\mu\in\mathcal{K}}\,\ell_t(\mu)$$
$$= \underbrace{\sum_{t=1}^{T}\mathbb{E}_{t-1}\left[\ell_t(\hat{\mu}_t) - \ell_t(\nu_{t+1})\right]}_{:=S_1} + \underbrace{\sum_{t=1}^{T}\mathbb{E}_{t-1}\left[\ell_t(\nu_{t+1}) - \ell_t(\mu_t^*)\right]}_{:=S_2}$$

The sum $S_1$ is controlled by applying Lemma E.3. We then obtain with $Y_t = \langle \nabla_t, \hat{\mu}_t - \nu_{t+1} \rangle$ with probability $1 - \delta$ :

$$S_1 \le \sum_{t=1}^{T} Y_t + \frac{\beta}{2} \sum_{t=1}^{T} Y_t^2 + \frac{\beta - \alpha}{2} \mathbb{E}_{t-1}[Y_t^2] + \frac{2}{\beta} \log(1/\delta).$$

The first sum is controlled by an intermediary result given in Lemma D.5, the second by Cauchy-Schwarz, we then have:

$$
\begin{aligned}
\sum_{t=1}^{T} Y_t &= \sum_{t=1}^{T} \langle \hat{\mu}_t - \nabla_t, \hat{\nu}_t \rangle + \langle \nabla_t, \hat{\nu}_t - \nu_{t+1} \rangle \\
&\le \frac{1}{2\gamma} \sum_{t=1}^{T} \nabla_t^\top A_t^{-1} \nabla_t + \frac{\gamma}{2} \sum_{t=1}^{T} (\hat{\mu}_t - \nu_t)^\top \nabla_t \nabla_t^\top (\hat{\mu}_t - \nu_t) + \frac{1}{2\gamma} + GP_T(\nu)
\end{aligned}
$$

Recall that, because $\gamma = \frac{1}{2} \min(\frac{1}{GD}, \alpha/4)$, $\frac{1}{\gamma} \le 2 \left( \frac{4}{\alpha} + GD \right)$, one has $\sum_{t=1}^{T} \nabla_t^\top A_t^{-1} \nabla_t \le 2 \left( \frac{8}{\alpha} + GD \right) d \log(T)$. Finally, one has:

$$\sum_{t=1}^{T} Y_t \le 2 \left( 1 + \frac{8}{\alpha} + GD \right) d \log(T) + \frac{\alpha}{16} \sum_{t=1}^{T} \left( \nabla_t^\top (\hat{\mu}_t - \nu_t) \right)^2 + GP_T(\nu)$$

Plus, remarking that:

$$
\begin{aligned}
\left( \nabla_t^\top (\hat{\mu}_t - \nu_t) \right)^2 &= \left( \nabla_t^\top (\hat{\mu}_t - \nu_{t+1}) + \nabla_t^\top (\nu_{t+1} - \nu_t) \right)^2 \le 2Y_t^2 + 2 \left( \nabla_t^\top (\nu_{t+1} - \nu_t) \right)^2 \\
&\le 2Y_t^2 + 2G^2 \|\nu_{t+1} - \nu_t\|^2
\end{aligned}
$$

Summing on $t$ and reorganising the previous bounds finally gives:

$$S_1 \le GP_T(\nu) + 2G^2 S_T(\nu) + \frac{\beta + \alpha/4}{2} \sum_{t=1}^{T} Y_t^2 + \frac{\beta - \alpha}{2} \mathbb{E}_{t-1}[Y_t^2] + \frac{2}{\beta} \log(1/\delta) + O(d \log(T))$$

Finally, because $|Y_t| \le GD$ a.s, we exploit a corollary of a Poissonian inequality stated in (Wintenberger, 2021, Eq. (7)). With probability $1 - \delta$ we have:

$$\sum_{t=1}^{T} Y_t^2 \le 2 \sum_{t=1}^{T} \mathbb{E}_{t-1}[Y_t^2] + 2(GD)^2 \log(1/\delta) \tag{8}$$

Thus, taking an union bound and $\beta$ such that $3\beta - \alpha/2 = 0$ gives us with probability $1 - 2\delta$:

$$S_1 \le O(d \log(T)) + GP_T(\nu) + G^2 S_T(\nu) + \left( \frac{12}{\alpha} + \frac{10\alpha}{24} (GD)^2 \right) \log(1/\delta)$$

Finally, to control $S_2$, we reuse the arguments provided in Thm. 3.3. More precisely, we use that the step size of CONSTRUCT allow us to use Lemma 2.5 to claim that for any $t \geq 0$:

$$\ell_t(\nu_{t+1}) - \ell_t(\mu_t^*) \leq \frac{G^2}{K} \sum_{j=1}^{K} \eta_j'$$

$$\leq \frac{G^2(1 + \log(K))}{\lambda K}$$

Then, because $K = T$, applying our conditional expectations and summing over $t$ gives us.

$$S_2 \leq \frac{G^2}{\lambda}(1 + \log(T)) = O(\log(T)).$$

Summing $S_1$ and $S_2$ concludes the proof.

$\square$

### E.4 Proof of Thm. 3.6

*Proof.* We first state that for any $(\hat{\mu}_t, \mu_t)$:

$$\sum_{t=1}^{T} \mathbb{E}_{t-1}[\ell_t(\hat{\mu}_t)] - \sum_{t=1}^{T} \mathbb{E}_{t-1}[\ell_t(\mu_t)] = \sum_{t=1}^{T} \mathbb{E}_{t-1}\left[\ell_t(\hat{\mu}_t) - \ell_t(\mu_t)\right]$$

$$\leq \sum_{t=1}^{T} \mathbb{E}_{t-1}\left[\ell_t(\hat{\mu}_t) - \ell_t(\mu_t^*)\right] \text{ with } \mu_t^* = \mathrm{argmin}_{\mu \in \mathcal{K}} \ell_t(\mu)$$

$$= \underbrace{\sum_{t=1}^{T} \mathbb{E}_{t-1}\left[\ell_t(\hat{\mu}_t) - \ell_t(\nu_{t+1})\right]}_{:=S_1} + \underbrace{\sum_{t=1}^{T} \mathbb{E}_{t-1}\left[\ell_t(\nu_{t+1}) - \ell_t(\mu_t^*)\right]}_{:=S_2}$$

The sum $S_1$ is controlled by applying Lemma E.3. We then obtain with $Y_t = \langle \nabla_t, \hat{\mu}_t - \nu_{t+1} \rangle$ with probability $1 - \delta$ :

$$S_1 \leq \sum_{t=1}^{T} Y_t + \frac{\beta}{2} \sum_{t=1}^{T} Y_t^2 + \frac{\beta - \alpha}{2} \mathbb{E}_{t-1}[Y_t^2] + \frac{2}{\beta} \log(1/\delta).$$

The first sum is controlled by an intermediary result given in proposition D.6, the second by Cauchy-Schwarz, we then have:

$$\sum_{t=1}^{T} Y_t = \sum_{t=1}^{T} \langle \hat{\mu}_t - \nabla_t, \hat{\nu}_t \rangle + \langle \nabla_t, \hat{\nu}_t - \nu_{t+1} \rangle$$

$$\leq \sqrt{2}D\left(1 + \sqrt{\min_{H \in \mathcal{H}} \sum_t \|\nabla_t\|_H^{*2}}\right) + GP_T(\nu)$$

Reorganising the previous bounds finally gives:

$$S_1 \leq GP_T(\nu) + \frac{\beta}{2}\sum_{t=1}^{T} Y_t^2 + \frac{\beta - \alpha}{2}\mathbb{E}_{t-1}[Y_t^2] + \frac{2}{\beta}\log(1/\delta)$$

Finally, because $|Y_t| \leq GD$ a.s, we exploit a corollary of a Poissonian inequality stated in (Wintenberger, 2021, Eq. (7)). With probability $1 - \delta$ we have:

$$\sum_{t=1}^{T} Y_t^2 \leq 2\sum_{t=1}^{T} \mathbb{E}_{t-1}[Y_t^2] + 2(GD)^2 \log(1/\delta) \tag{9}$$

Thus, taking an union bound and $\beta$ such that $3\beta - \alpha = 0$ gives us with probability $1 - 2\delta$:

$$S_1 \leq \sqrt{2}D\left(1 + \sqrt{\min_{H \in \mathcal{H}}\sum_t \|\nabla_t\|_H^{*2}}\right) + GP_T(\nu) + \left(\frac{2}{\alpha} + \frac{2\alpha}{3}(GD)^2\right)\log(1/\delta)$$

Finally, to control $S_2$, we reuse the arguments provided in Thm. 3.3. More precisely, we use that the step size of CONSTRUCT allow us to use Lemma 2.5 to claim that for any $t \geq 0$:

$$\ell_t(\nu_{t+1}) - \ell_t(\mu_t^*) \leq \frac{G^2}{K}\sum_{j=1}^{K}\eta_j'$$

$$\leq \frac{G^2(1 + \log(K))}{\lambda K}$$

Then, because $K = T$, applying our conditional expectations and summing over $t$ gives us.

$$S_2 \leq \frac{G^2}{\lambda}(1 + \log(T)) = O(\log(T)).$$

Summing $S_1$ and $S_2$ concludes the proof.

$$\square$$

