# OpenReview forum: "Optimistic Dynamic Regret Bounds"
_TMLR — Rejected by TMLR_

### Review · Reviewer_SAPa · 2023-02-02

**Summary Of Contributions:**

This paper considers online learning problems with strongly-convex loss functions and focuses on dynamic regret, which is the difference between cumulative losses for the algorithm and for the optimal output sequence.
A main contribution of this paper is to propose a new online learning algorithm that exploits experts' advice, in which the framework of optimistic online learning, e.g., by Rakhlin and Sridharan (2013a;b), is followed.
In addition, an algorithm for constructing experts' advice is presented, which leads to dynamic-regret bounds dependent on the path-length of the experts' advice.

**Audience:**

Yes

**Broader Impact Concerns:**

I have no concerns about broader imact.

**Claims And Evidence:**

No

**Requested Changes:**

I would like you to address the issues I pointed out above.

**Strengths And Weaknesses:**

Strength
- The paper is well organized and the algorithms and theory are clearly presented.

Weakness
- The correctness of proofs of some theorems is somewhat questionable.
In Theorem 3.2, step size $\eta'$ for Algorithm 2 is defined by $\eta'_j = \frac{D}{\lambda G j}$.
Under this definition, the assumption of $1/\eta'\_j - \lambda \le 1 / \eta'\_{j-1}$ imposed in Lemma 2.5 is not always satisfied.
This point should be noted in the evaluation of $S_3$ in the proof of Theorem 3.2 (page 27), but in the present manuscript it is only asserted that $S_3 = \tilde{O}(\sqrt{T})$ without any evidence.
- There are some unclear points regarding the results of comparison with existing studies.
For example, I would like to verify the meaning of the following statements given just before Theorem 3.2:
"Furthermore, notice that our convergence rate is lower than the $O(min(P_T , S_T ))$ rate provided in Zhang et al. (2017),"
My understanding is that in most existing studies, including (Zhang et al., 2017), dynamic-regret bounds depending on path-lengths $P_T(\mu^*), S_T(\mu^*)$ for the \textit{true optimal solution sequences} $\mu^*$ are provided.
On the other hand, the regret upper bounds proven in this paper depends on path-lengths $P_T(\nu), S_T(\nu)$ for \textit{exprets' advice sequcences} $\nu$.
In general, the values of $(P_T(\mu^*), S_T(\mu^*))$ and $(P_T(\nu), S_T(\nu))$ are different.
Any claims about the relationship between $(P_T(\mu^*), S_T(\mu^*))$ and $(P_T(\nu), S_T(\nu))$ (e.g., inequality relations) could not be found in the paper.
This means that we cannot say that the proposed algorithm is sperior to existing results.
- There are many typos and unclear defintions:
	- page 4: definiton of $\mathbb{N}^*$ cannot be found.
	- $\|| \cdot \||$: It would be better to spesify what kind of norm it is. (I guess this represents the Eucledian norm)
	- Definition 2.1: Although $\mu\_{temp}$ is intorduced in the text, $\mu$ is used in the formula.
	- Definition 2.1: $\mu\_{temp} = ( \hat{\mu}\_{temp,t} )\_{t\ge 1 )}$ closing parenthesis at the end would be redundant.
	- Lemma 2.3: definite positive $H$ <- positive definite $H$?
	- Algorithms 3, 4:  Epoch T <- Epoch $T$
	- Theorem 3.1: (Algroithm 3 ) <- (Algorithm 3)   (There is extra space.)
	- Theorem 3.1: $(\frac{1}{\lambda j})\_{j=(1..K)}$ <- $(\frac{1}{\lambda j})\_{j = 1..K}$ ?
	- Two styles for representing sequences are mixed. $(\cdot)_{j=1..K}$ vs $(\cdot)_{j=(1..K)}$
	- Paragraph after Theorem 3.1: extra space before Zinkevich, 2003
	- Theorem 3.2: Redundant closing parenthesis after $\eta = ...$
	- Theorem 3.2: Space after $T\geq 1$,...
	- Paragraph after tere 3.3 $min(P_T, S_T)$ <- $\min(P_T, S_T)$
	- Second paragraph of Page 2: $O(\min(P^*_T, S^*_T)$ <- $O(\min(P^*_T, S^*_T))$
	- Second paragraph of Page 3: $(E_{t-1}[\ell\_t])\_{t \ge 1}=E[\ell\_t | F\_{t-1}]$ <- $E\_{t-1}[\ell\_t] = E[\ell\_t | F\_{t-1}]$ ?
	- Theorem B.1: $\sum_{t=1} \eta_t$ <- $\sum_{t=1}^T \eta_t$ (similar typos can be found in pages 17, 18, 19, 22, 23...)
	- Definition D.1 $\hat{\mu_t}$ <- $\hat{\mu}_t$ (similar ones can be found in pages 26, 27, ...)
- Some proofs in the appendix looks imcomplete.
For example, the proof of Theorem B.1 includes "We use the following lemma: [nothing] We then have: [formula] Hence: [formula] So: [Formula]"

---

> ### Author Response · Authors · 2023-02-10
> **Thank you for your review, we corrected all the typos.**
>
> We thank you for your careful review of our work which allowed us to detect many typos and approximations. We are happy to read that you find our paper well organised, and we adress below your various point of inquiry.
>
> - (About Theorem 3.2) We thank you for spotting this error in the statement of Theorem 3.2. Indeed, the right step sizes required to apply CONSTRUCT are $\mathbf\{\eta'\}=(\frac\{1\}\{\lambda j\})_\{j=(1..K)\}$, again with $K= \lceil \sqrt\{T\} \rceil$.  This allows to apply properly Lemma 2.5 and so, Thm 3.2 holds. Thanks to your remark, we spotted similar ersatzs in both Theorems 3.4 and 3.6. We corrected all theorem statements and detailed all the associated proofs from p.27 to 31, all changes are written in red.
> - (Comparison with literature) In general, the path length of any expert sequence $\nu$ is not directly comparable with the minimisers $\mu^*$. However, note that our theorems hold when for any $t, \nu_\{t+1\} = \mu_t^*$ (as precised in the last sentence of all of our theorems, this corresponds to the case where $S_3=0$ in Theorem 3.2's proof for instance.). In this case we have the equality $P_T(\nu)= P_{T-1}(\mu^*) + ||\mu_1^* - \nu_1||$. We are then able to compare our convergence rate to those of the literature.
>
>   Furthermore, we precise that taking $\nu_{t+1} = \mu_t^*$ corresponds to the case where the environment gives to the learner at time $t+1$ the best possible predictor at time $t$. This is not always possible as this minimiser may have no close form (hence the interest of \textsc{Construct} to approximate it). To us, an interest of our results is that we can upper bound the regret of our algorithm with respect to the true minimisers $\hat{\mu}^*$ by the empirical paths $P_T(\nu),S_T(\nu)$. This allow us to obtain a computable bound even when the minimisers are not revealed to the learner.
>   We detailed this point in red right above Thm 3.2.
>
>   Finally even when comparison is possible, the term 'lower' (used to describe our convergence rate) is ill-chosen. Indeed, what we wanted to say is that our convergence rate is worse than the one of Zhang et al.,2017. However our result is more general as we did not assume our losses to be smooth. We rectified this point in the main document.
> - Typos:
>    - $\mathbb{N}^*$ denotes $\mathbb{N}/ \\{0\\}$ we replaced all the occurences of this notation.
>    - $||.||$ denotes indeed, the euclidean norm, we precised this in red page 4.
>    - We corrected all other typos you mentionned, we thank you for your time.
> - (End of Theorem B.1's proof) We thank you for spotting this, the sentence about the following lemma is not supposed to exist. For the sake of completeness we detailed more carefully the end of the proof. Do you have supplementary concerns about other proofs of the paper?

---

### Review · Reviewer_2CqF · 2023-02-05

**Summary Of Contributions:**

This paper provides analysis of dynamic regret algorithms with respect to the sequence of individual loss minimizers $\mu^\star_t$ of a sequence of strongly convex and Lipschitz losses $\ell_t$. The algorithms are based upon a two-stage procedure: first, one uses a “CONSTRUCT” method (which is simply gradient descent in the paper, but could in principle be any convex optimization algorithm) to find an approximation of the minimizer of the current loss $\ell_t$. Then, one uses an extra procedure “ADJUST” to “fix” the iterates of some other static-regret algorithm to be “more like” the outputs of the CONSTRUCT procedure in certain cases. The analysis proceeds by bounding the difference between the ADJUST and CONSTRUCT outputs as well as the performance of the CONSTRUCT output. A key improvement is that in contrast to the prior work of Zhang et al 2017, no smoothness assumptions are made in this work.


**Audience:**

No

**Claims And Evidence:**

Yes

**Requested Changes:**

Please explain if and how the ADJUST procedure improves upon simply returning $\nu$. This is critical for acceptance. Please also address the other issues raised in the review.


**Strengths And Weaknesses:**


When reading this paper, I found myself very confused about what the advantages of the “ADJUST” approach are. As far as I can see, ADJUST is the primary technical innovation here: the CONSTRUCT algorithm appears to be the same trick as employed by Zhang et al 2017.


However, it appears to me that I can strictly improve upon the result of Theorem 3.1 by simply setting $\hat \mu_{t+1} = \nu_{t+1}$ without appealing to ADJUST at all (essentially, instead of running the ADJUST algorithm as proposed in the paper, simply have ADJUST immediately return its input $\nu$). Then, the regret would be bounded by:
$$
\sum_{t=1}^T \ell_t(\nu_t) -\ell_t(\nu_{t+1}) + \sum_{t=1}^T \ell_t(\nu_{t+1}) -\ell_t(\mu^\star_t)
$$
Which is basically equation (1) in the appendix without the first summation. Then, the first of these summations is clearly at most $\sum_{t=1}^T \langle \nabla \ell_t(\nu_t), \nu_t-\nu_{t+1}\rangle - \frac{\lambda}{2}\|\nu_t-\nu_{t+1}\|^2\le GP(\nu) - \lambda S(\nu)$, and the second term is bounded by $\tilde O(G^2T/K\lambda)=\tilde O(G^2\sqrt{T}/\lambda)$ whenever $K=\sqrt{T}$. This appears to remove the $GD\sqrt{T}$ term from Theorem 3.1 and so is strictly better.

The same issue is present for Theorem 3.2.

In section 3.2, are we changing assumptions? Are the losses no longer strongly convex? The theorem statements seem to continue to imply that the losses are strongly convex, but if so then the results of Theorem 3.3 and 3.4 seem strictly weaker than 3.1 and 3.2 - worse regret and a slower algorithm. Why not just use those theorems? If the losses are no longer strongly convex, what is the $\lambda$ parameter in these theorems?

Note that in contrast to what appears to be stated in the paragraph after Theorem 3.3, as far as I  can tell, the previous theorem compared to here (Zhang et al 2017 Theorem 11) does NOT seem to assume strong convexity. Instead, this prior result assumes self-concordance, which is rather different.

Nevertheless, even if we relax the assumptions to self-concordance, clearly one could with sufficient computation (e.g. a newton-step) ensure that the “additional knowledge” sequence $\nu$ has arbitrarily small regret. Then, just as in Theorem 3.1 and 3.2 I do not see why I should not just omit the ADJUST procedure and simply set $\hat \mu_{t+1} = \nu_{t+1}$.

The same issue is also present for Theorem 3.5 and 3.6: why attempt to use a more advanced algorithm for choosing $\hat \mu_{t+1}$ when we are just going to compare it to $\nu_{t+1}$, which is already known? It still seems we should set $\hat \mu_{t+1}=\nu_{t+1}$.

Beyond these issues, there is another problem: the regret bounds appear to be in terms of the variation of $\nu$, rather than the variation of the comparator sequence $\mu_\star$. This makes them very hard to interpret. Instead, I feel one should increase the accuracy of the CONSTRUCT procedure so that one can ensure $P(\nu)\approx P(\mu_\star)$.

Finally, note that the regret bound for the full-matrix adagrad algorithm is actually strictly worse by a factor of up to $\sqrt{d}$ than the regret bound for simple online gradient descent with the “scalar adaptive” learning rate $\eta_t = 1/\sqrt{\sum_{i=1}^ \|g_i\|^2}$. See http://blog.wouterkoolen.info/GrokkingAdaGrad/post.html or https://proceedings.neurips.cc/paper/2020/hash/6495cf7ca745a9443508b86951b8e33a-Abstract.html for more detail about this. Thus, even if these other issues can be addressed, I would recommend using a different base algorithm.

More minor presentation issues:

Definition 2.1 is $\hat \mu$ the same as $\hat \mu_{temp}$? Otherwise $\hat \mu_{temp}$ does not appear to be used in the definition… Further, what is a “sequence of additional knowledge”? The phrase “additional knowledge” is an imprecise english term. Please precisely define what is meant by this so that in later contexts it is clear what is meant. My belief from reading further is that an “additional knowledge” is simply a vector in the same space as $\hat \mu$, but this should be made clear *before* the term is used. For example , one could say instead “a sequence of additional knowledge $\nu$ in the form of vectors in $\mathbb{R}^d$...” for example.

Paragraph before algorithm 1: “We precise that “ doesn’t make sense. I guess you mean to define something here as in “We denote by..”

Many of the sentences either don’t make sense or are hard to read. Part of this is simply a few minor grammatical errors that can usually be parsed around, but others are due to use of what appear to be undefined technical terms, which is a more serious issue. For example, in the paragraph after algorithm 1, the phrase: “it means that in the referential centered in $m_t, \hat \mu_{temp, t+1}$ does not point in the same direction than the dynamic $\nu_{t+1} − \nu_t$” uses the undefined terms “referential” and “dynamic”. Please carefully check for these issues and define all relevant terms as it makes the writing very confusing. In some cases I suspect that these undefined terms should actually just be deleted - in the example quoted here for instance my best interpretation is that the sentence should have read: “it means that $m_t, \hat \mu_{temp, t+1}$ does not point in the same direction as $\nu_{t+1} − \nu_t$”

Please also run a spell-checker, there are few small errors (“aditionnal” in the title for section 2.2, “aproximation” later on in the section etc).

---

> ### Author Response · Authors · 2023-02-10
> **Thank you for your review, answer to your main concern**
>
> We thank you for your detailed review which will help a lot to enhance our work. regarding your main concern, we provide below a sum up of the clarifications and discussions we added in the (new) Appendix B of our work which is in written in red, which explains in detail the place of this contribution within the optimistic online learning literature.
> We focus here on the D-OGD algorithm (associated to Theorems 3.1 and 3.2)
>
> **A general framework for OGD,OMGD, D-OGD**
>
> In the optimistic framework, we are given experts advices materialised as additional knowledge $\nu$ being a sequence of vectors in $\mathbb{R}^d$.
> We then define an *optimistic gradient-based (OGB) algorithm with judge* $f$ as an online algorithm satisfying the following pattern:
> At time $t$,
>
> -Perform a gradient descent step to obtain $\hat{\mu}_{temp,t+1}$.
>
> -Apply a judge $f$ using additional knowledge $\nu$: $\hat\{\mu\}_\{t+1\}= f(t,\hat\{\mu\}_\{temp,t+1\},\nu)$.
>
> The role of the judge $f$ within an OGB algorithm is to determine how we combine a given additional knowledge with a classical gradient descent. The choice of the judge depends on the confidence we have on the quality of the additional knowledge $\nu$.
>
> We now assume here that our additional knowledge is given by \textsc{Construct}. In this case both OMGD and OGD are optimistic gradient-based. Indeed:
> - OMGD is an OGB algorithm with judge $f(t,\nu,\hat\{\mu\}_\{temp,t+1\}) = \nu_\{t+1\}$. This corresponds to the case where the judge estimate that the additional knowledge is perfectly relevant for the next prediction.
> - OGD is an OGB algorithm with judge $f(t,\nu,\hat\{\mu\}_\{temp,t+1\}) = \hat\{\mu\}_\{temp,t+1\}$. This corresponds to the case where the judge estimates that the additional knowledge is useless or adversarial, it will then choose to ignore it.
> - Between those two extremes lies D-OGD which is an OGB algorithm with judge $f=$ADJUST. This corresponds to the case of a moderate judge which find a balance between the impact of the gradient descent step (an exploration phase) and the one of the additional knowledge (an exploitation one)
>
>
> The case you pointed out ($\hat\{\mu\}_\{t+1\}= \nu_\{t+1\}$) corresponds to OMGD. This tightens our bound for strongly convex losses to the formula you wrote on the second paragraph of the 'Strength and Weaknesses' part. However this corresponds to a naive strategy as it implies an absolute confidence on $\nu$. This may be suboptimal in practice as shown in Section 4.2. Our work aims to provide convergence guarantees for more elaborated strategies (from an optimistic point of view).
>
> **D-OGD as a flexible optimistic algorithm with dynamic regret bounds.**
>
> As noticed above, it is possible to recover OMGD and OGD as OBD algorithms whose judges choose either to fully trust or ignore the additional knowledge. Note that those judges do not truly combine additional knowledge and the output of a gradient descent step: its either choosing one or the other. However, optimistic online learning aims to combine those quantities: this is for instance what proposes Rakhlin et al. 2013s Optimistic Mirror Descent which has static regret bounds.
>
> A legitimate question is then: is it possible to design an optimistic online algorithm which perpetrates the way Optimistic Mirror Descent deals with additional knowledge while achieving dynamic regret bounds?
>
>
> The research field behind this question is not new as it has been already investigated for instance by Jadbabaie et al. 2015. They succeeded to provide an adaptive version of OMD (at the cost of a Lipschitz assumption for the Bregman divergence) when the additional knowledge is given on the gradient space.
>
> We contribute to this line of work through three optimistic algorithms based on ADJUST all valid for strongly convex functions. We succeed to maintain theoretical guarantees for our optimistic strategy, but this comes at the cost of a deteriorated convergence rate with respect to the naive optimistic strategies lying behind OMGD and OGD.
>
> We insist on the fact that the interest of ADJUST is NOT to tighten existing bounds. Instead, we provide algorithms with non-naive optimistic strategies holding for an additional knowledge lying on the predictor space. We also furnish convergence guarantees and leaves the question of the tightness as an open problem for now. Note also that our algorithms can be applied with any additional knowledge satisfying a technical assumption: our approach encompasses OMGD and OGD as particular cases.
>
> We also provide an experimental setup where D-OGD performs better than OGD and OMGD: the Online Quadratic Problem of Sec 4.2. This situation exhibits a noisy problem, where trusting totally our experts (OMGD) or ignoring them totally (OGD) leads to poorer cumulative risks than D-OGD. This illustrates the relevance of our method to exploit efficiently additional knowledge.
>
> We hope this shed a new light on the interest of our work.

---

> > ### Author Response · Authors · 2023-02-10
> > **Answer to your other concerns**
> >
> > **Other points of the review**
> >
> > - We are not sure to understand your remark about the vacuousness of D-ONS guarantees. For instance Theorem 3.3 provides an upper bound of $GP_T(\nu) - \lambda S_T(\nu) + \mathcal{O}\left(d\log(T)   \right)$ while Theorem 3.1 propose the rate $GP_T(\nu) - \lambda S_T(\nu) + \mathcal{O}\left(\log(T)\sqrt{T}   \right)$ which is strictly greater.
> > - It is true that (Zhang et al. 2017, Theorem 3.11) relies only on the self concordance assumption. Thank you for pointing this out, we modified the associated paragraph in red.
> > - We interpret the path lengths $P_T(\nu),S_T(\nu)$ as adaptations of $P_T(\mu^*), S_t(\mu^*)$ to the optimistic online learning framework. Indeed, consider the case where at time $t+1$, the previous true minimiser $\mu_t^*$ is revealed to the learner. Then taking $\nu_{t+1}= \mu_t^*$ allow us have $P_T(\nu)= P_{T-1}(\mu^*) + ||\nu_1- \nu_0||$ (similar result for $S_T(\nu)$). On the more general situation where only the loss function is revealed (and not the true minimiser directly), we need to approximate it (e.g. through \textsc{Construct}). And then $P_T(\nu)$ appears as an empirical surrogate of $P_T(\mu^*)$.
> > - Thank you for pointing out the reference about the non-tightness of Full-Adagrad regret bounds. We note that the work from Cutkosky holds for static regret bounds while we provide dynamic ones. Is there an extention of Cutkosky's work to our setting? If none, given the wide use of Adagrad in practice, we believe it remains worth to mention our dynamic regret bounds stated in theorems 3.5 and 3.6.
> > - We corrected in red the definition 2.1 to get rid of $\mu$. We also precised in red in Section 2.1 that additional knowledge was a sequence of vectors of $\mathbb{R}^d$.
> > - We corrected and simplified the approximations you find in our work, we thank you for pointing this out.

---

### Review · Reviewer_udFd · 2023-03-01

**Summary Of Contributions:**

The paper studies dynamic regret in online convex optimization with strongly convex losses, where the regret is defined as the learner's total loss subtract the optimal sequence's total loss.  The dynamic regret is quantified by standard quantities such as $D$ (diameter of the feasible), $G$ (gradient norm), and $\lambda$ (the strongly convex parameter), as well as two quantities that characterize the amount of change in the environment: $P_T(\nu), S_T(\nu)$, which are first-order and second-order path-length, respectively. The $\nu_t$ here is some "expert's advice" which is supposed to be closed to the minimizer of $\ell_{t-1}$. Through the paper, it is set to the output of Algorithm 2, which is essentially a gradient descent algorithm over the function $\ell_{t-1}$. The paper proposes a new procedure called "Adjust" to adjust the update direction of standard algorithms if the update direction is not aligned with the move of $\nu_t$. Three algorithms Dynamic Online Gradient Descent (D-OGD), Dynamic Online Newton Step (D-ONS), and Dynamic Adagrad (D-Adagrad) are proposed, all using Adjust as a subroutine. The dynamic regret bounds are $GP_T(\nu) - \lambda S_T(\nu)$ plus standard regret bound for OGD, ONS and Adagrad respectively.

The contribution is unclear to me. While the authors claim that previous work by [Zhang et al. 2017] achieves a better bound $\min(P_T(\mu^\star), S_T(\mu^\star))$ under an additional assumption of smoothness, it is proven in the Theorem 3 of [Zhao and Zhang, 2021] that a bound of $GP_T(\mu^\star)$ is achievable without the smoothness assumption. In their analysis, the negative term $-\lambda S_T(\mu^\star)$ is omitted just for simplicity, so they actually achieve $GP_T(\mu^\star) - \lambda S_T(\mu^\star)$, which is already better than the bound obtained by the authors (because Zhao and Zhang don't have additional term coming from OGD, ONS, Adagrad), also without making stronger assumptions. In terms of algorithm, theirs is a simple greedy algorithm, while the one proposed here involves much more parts which don't seem to be necessary.

Overall, I think the main contribution needs to be clarified in the first place before deciding whether it can be accepted or not.


[Zhang et al., 2017] Improved Dynamic Regret for Non-degenerate Functions

[Zhao and Zhang, 2021] Improved Analysis for Dynamic Regret of Strongly Convex and Smooth Functions (https://arxiv.org/pdf/2006.05876.pdf)



**Audience:**

No

**Claims And Evidence:**

No

**Requested Changes:**

See the "Summary of Contribution" part.

**Strengths And Weaknesses:**

See the "Summary of Contribution" part.

---

> ### Author Response · Authors · 2023-03-02
> **Thank you for your review, more precisions below on our contribution.**
>
> We thank you for your review. We realise that the specificities of our contribution were unclear in the first version of our work as noticed by you and Reviewer 2CqF. We provided in the new appendix B (in red) of our work more informations about the place of our contribution within the optimistic online learning literature.
> Similarly to our answer to Rev. 2CqF, we provide below a general framework in which the place of our contributions becomes clearer.
>
> **A general framework for OGD,OMGD, D-OGD**
>
> In the optimistic framework, we are given experts advices materialised as additional knowledge $\nu$ being a sequence of vectors in $\mathbb{R}^d$.
> We then define an *optimistic gradient-based (OGB) algorithm with judge* $f$ as an online algorithm satisfying the following pattern:
> At time $t$,
>
> -Perform a gradient descent step to obtain $\hat{\mu}_{temp,t+1}$.
>
> -Apply a judge $f$ using additional knowledge $\nu$: $\hat\{\mu\}_\{t+1\}= f(t,\hat\{\mu\}_\{temp,t+1\},\nu)$.
>
> The role of the judge $f$ within an OGB algorithm is to determine how we combine a given additional knowledge with a classical gradient descent. The choice of the judge depends on the confidence we have on the quality of the additional knowledge $\nu$.
>
> We now assume here that our additional knowledge is given by \textsc{Construct}. In this case both OMGD and OGD are optimistic gradient-based. Indeed:
> - OMGD is an OGB algorithm with judge $f(t,\nu,\hat\{\mu\}_\{temp,t+1\}) = \nu_\{t+1\}$. This corresponds to the case where the judge estimate that the additional knowledge is perfectly relevant for the next prediction.
> - OGD is an OGB algorithm with judge $f(t,\nu,\hat\{\mu\}_\{temp,t+1\}) = \hat\{\mu\}_\{temp,t+1\}$. This corresponds to the case where the judge estimates that the additional knowledge is useless or adversarial, it will then choose to ignore it.
> - Between those two extremes lies D-OGD which is an OGB algorithm with judge $f=$ADJUST. This corresponds to the case of a moderate judge which find a balance between the impact of the gradient descent step (an exploration phase) and the one of the additional knowledge (an exploitation one)
>
> The case you pointed out ($\hat\{\mu\}_\{t+1\}= \nu_\{t+1\}$) corresponds to OMGD. This tightens our bound for strongly convex losses to the formula you wrote on the second paragraph of the 'Strength and Weaknesses' part. However this corresponds to a naive strategy as it implies an absolute confidence on $\nu$. This may be suboptimal in practice as shown in Section 4.2. Our work aims to provide convergence guarantees for more elaborated strategies (from an optimistic point of view).
>
> **D-OGD as a flexible optimistic algorithm with dynamic regret bounds.**
>
> As noticed above, it is possible to recover OMGD and OGD as OBD algorithms whose judges choose either to fully trust or ignore the additional knowledge. Note that those judges do not truly combine additional knowledge and the output of a gradient descent step: its either choosing one or the other. However, optimistic online learning aims to combine those quantities: this is for instance what proposes Rakhlin et al. 2013s Optimistic Mirror Descent which has static regret bounds.
>
> A legitimate question is then: is it possible to design an optimistic online algorithm which perpetrates the way Optimistic Mirror Descent deals with additional knowledge while achieving dynamic regret bounds?
>
>
> The research field behind this question is not new as it has been already investigated for instance by Jadbabaie et al. 2015. They succeeded to provide an adaptive version of OMD (at the cost of a Lipschitz assumption for the Bregman divergence) when the additional knowledge is given on the gradient space.
>
> We contribute to this line of work through three optimistic algorithms based on ADJUST all valid for strongly convex functions. We succeed to maintain theoretical guarantees for our optimistic strategy, but this comes at the cost of a deteriorated convergence rate with respect to the naive optimistic strategies lying behind OMGD and OGD.
>
> We insist on the fact that the interest of ADJUST is NOT to tighten existing bounds. Instead, we provide algorithms with non-naive optimistic strategies holding for an additional knowledge lying on the predictor space. We also furnish convergence guarantees and leaves the question of the tightness as an open problem for now. Note also that our algorithms can be applied with any additional knowledge satisfying a technical assumption: our approach encompasses OMGD and OGD as particular cases.
>
> We also provide an experimental setup where D-OGD performs better than OGD and OMGD: the Online Quadratic Problem of Sec 4.2. This situation exhibits a noisy problem, where trusting totally our experts (OMGD) or ignoring them totally (OGD) leads to poorer cumulative risks than D-OGD. This illustrates the relevance of our method to exploit efficiently additional knowledge.
>
> We hope this shed a new light on the interest of our work.

---

> > ### Comment · Reviewer_udFd · 2023-05-02
> > **Additional comments**
> >
> > I still feel that usefulness of the techniques developed in this paper is unclear. First, though the authors claim that the techniques is general, they did not provide a convincing example where the incorporation of judge, ADJUST etc. improves the regret; instead, the regret is deteriorated for the simplest setting as pointed out in the original review. Second, although the authors provides several choices for the judge function, it's unclear how to pick from them when facing an unknown environment. It seems that picking the wrong one can worsen the performance. Overall, I feel that the work did not really complete the story, providing a principled way to choose the parameters in the algorithm, and demonstrating that the algorithm is robust to false choice.
> >
> > I would like to point out again that the comparison with Zhang et al. (2017) in Page 7 is misleading, because Zhang et al. (2017) actually do not require the additional smoothness assumption --- please see my original review for details.

---

### Decision · Action_Editors · 2023-05-26

**Recommendation:** Reject

**Comment:**

Given the reviews and the above comments, I believe the rationale behind this decision is clear. Another comment is that Reviewer udFd mentioned that a result of Zhang et al. (2017) could be improved by Zhao and Zhang (2021) so as to not require smoothness. I do not believe the authors responded to this point, and it is an important one. Overall, I advise the authors to justify the additional complexity of their algorithmic framework by clearly showing the benefit over past work. This may require careful comparisons, but overall this will allow the work to be appreciated upon publication.

**Audience:**

Related to the the goal of this work not being clear, I cannot at this time be confident that there would be sufficient audience for this work. The work appears to be targeted at the online learning theory community, but the (very well-informed) reviewers and also myself cannot presently see the advantage of the authors' new algorithms/algorithmic framework over previous work, and as such, I do not anticipate that there will be sufficient interest in this work in its present form.

**Claims And Evidence:**

As mentioned by Reviewer SAPa, there are many issues with the technical presentation, to the level that this work would require a major revision (outside of this review process) prior to having a clear evaluation of the claims and evidence. Even so, several reviewers were unsure of what innovations the authors are actually claiming. For instance, the authors mention that their goal is not to tighten existing bounds, and Reviewers udFd and 2CqF are in agreement that if this is not the goal, then it is not clear what the goal of this work is. Going forward, I urge the authors to clearly position this paper so that reviewers from learning theory can properly determined the advancements of this work.

**Resubmission Of Major Revision:**

The authors may consider submitting a major revision at a later time.